ecology

*Anopheles*, temperature, malaria, transmission, life history, senescence

**Author for correspondence:**
C. C. Murdock
e-mail: cmurdock@uga.edu

# Age influences the thermal suitability of *Plasmodium falciparum* transmission in the Asian malaria vector *Anopheles stephensi*

K. L. Miazgowicz[1,2,3], M. S. Shocket[7,8], S. J. Ryan[9,10,11], O. C. Villena[12], R. J. Hall[1,3,4], J. Owen[1], T. Adanlawo[4], K. Balaji[1], L. R. Johnson[12], E. A. Mordecai[7] and C. C. Murdock[1,2,3,4,5,6,13]

[1]Department of Infectious Diseases, College of Veterinary Medicine, [2]Center for Tropical and Emerging Global Diseases, [3]Center of Ecology of Infectious Diseases, [4]Odum School of Ecology, [5]Center for Vaccines and Immunology, and [6]River Basin Center, University of Georgia, Athens, GA, USA
[7]Biology Department, Stanford University, Stanford, CA, USA
[8]Department of Ecology and Evolutionary Biology, University of California, Los Angeles, CA, USA
[9]Quantitative Disease Ecology and Conservation (QDEC) Lab, Department of Geography, and [10]Emerging Pathogens Institute, University of Florida, Gainesville, FL, USA
[11]School of Life Sciences, University of KwaZulu-Natal, Durban, South Africa
[12]Computational Modeling and Data Analytics, Department of Statistics, Virginia Tech, Blacksburg, VA, USA
[13]Department of Entomology, Cornell University, Ithaca, NY, USA

SJR, 0000-0002-4308-6321; EAM, 0000-0002-4402-5547; CCM, 0000-0001-5966-1514

Models predicting disease transmission are vital tools for long-term planning of malaria reduction efforts, particularly for mitigating impacts of climate change. We compared temperature-dependent malaria transmission models when mosquito life-history traits were estimated from a truncated portion of the lifespan (a common practice) versus traits measured across the full lifespan. We conducted an experiment on adult female *Anopheles stephensi*, the Asian urban malaria mosquito, to generate daily per capita values for mortality, egg production and biting rate at six constant temperatures. Both temperature and age significantly affected trait values. Further, we found quantitative and qualitative differences between temperature–trait relationships estimated from truncated data versus observed lifetime values. Incorporating these temperature–trait relationships into an expression governing the thermal suitability of transmission, relative $R_0(T)$, resulted in minor differences in the breadth of suitable temperatures for *Plasmodium falciparum* transmission between the two models constructed from only *An. stephensi* trait data. However, we found a substantial increase in thermal niche breadth compared with a previously published model consisting of trait data from multiple *Anopheles* mosquito species. Overall, this work highlights the importance of considering how mosquito trait values vary with mosquito age and mosquito species when generating temperature-based suitability predictions of transmission.

## 1. Introduction

Despite the progress of global malaria elimination programs in reducing the incidence of human malaria, particularly *Plasmodium falciparum*, malaria remains a leading cause of infectious disease morbidity and mortality [1]. The occurrence of multi-class drug and insecticide resistance [1–3], in addition to alterations in mosquito behaviour [4], challenge our ability to eradicate malaria. While numerous factors affect the distribution and prevalence of mosquito-borne diseases, temperature is one of the most pervasive abiotic factors affecting both mosquito and pathogen vital rates [5]. Although the importance of these factors is increasingly recognized, gaps remain in the

current mechanistic understanding of the relationship between malaria risk and key environmental variables. Improving our understanding of the link between temperature and malaria transmission will be crucial for predicting how transmission varies geographically, seasonally, and with climate and land use change [6–9].

Recent research has begun to define the relationship between temperature and vector and pathogen traits relevant to transmission across a diversity of vector-borne disease systems [6,7,10–16]. The net effect of these traits on temperature-dependent transmission can be described by the basic reproduction number ($R_0$), defined as the number of secondary cases arising from a primary case introduced into a fully susceptible population. Transmission models that define $R_0$ can be used to generate predictions of disease risk and evaluate the efficacy of various interventions [17–20]. Key biological insights from previous models are that: (i) some regions of the world that are currently permissive for transmission may become less environmentally suitable as the climate warms past thermal optima and upper limits for malaria transmission; and (ii) vector control may become more difficult in northern latitudes as temperatures there become more permissive and suitable seasons extend [6,11,21].

Despite these advances, insights from previous expressions for $R_0$ derived from mechanistic models remain constrained by a lack of entomological and parasite data [10]. Temperature–trait relationships for key parameters are often indirectly estimated from a limited number of studies, leading to high uncertainty around the predicted thermal limits in current malaria $R_0$ formulations [10,11]. Additionally, the parameterization of $R_0$ expressions with temperature–trait relationships aggregated from different mosquito and parasite species likely introduces error and uncertainty into $R_0$ estimates due to variation in life history [10,11,22].

Further, evidence from a diversity of invertebrates demonstrates that organisms experience age-related changes in life-history traits [23–26]. These changes reflect either senescence, a decline in general physiological function with age, or a shift in resource allocation to different life-history tasks as an organism ages. Limited studies suggest that age modifies mosquito life history, with some evidence of reproductive senescence [27], alterations in biting frequency with age [28] and age-dependent survivorship [25]. Yet, we lack models for malaria that incorporate the combined effect of temperature and age on mosquito life-history traits. Often data are collected over a relatively limited portion of the mosquito lifespan and then used to estimate lifetime traits in models of mosquito population dynamics and disease transmission [6,7,10,11,14,15,29]. If key mosquito life-history traits vary with age, and temperature influences age-related changes in these traits, then the timing of when these traits are measured during the lifespan of the mosquito could impact the predicted relationships between these traits and temperature as well as the predicted thermal suitability for malaria transmission.

In this study, we conducted a life table experiment on the urban Indian malaria vector (*Anopheles stephensi*) at different constant temperatures. From these data, we calculated key life-history traits (i.e. lifespan, egg production and biting rates) in two different ways: first, we directly observed the trait values over the entire mosquito lifespan ('observed'); second, we estimated trait values from a truncated portion of the mosquito lifespan (as is typically done; 'estimated'). We used these trait data to answer the following questions.

(i) How do *An. stephensi* life-history traits vary across the full spectrum of biologically relevant temperatures? (ii) Do life-history traits that drive human malaria transmission vary with mosquito age? If so, (iii) do age-dependent changes in life history affect temperature–trait responses and the overall temperature suitability for transmission? And (iv) does the thermal response of transmission suitability change when traits from a single mosquito and parasite species are used (here), rather than aggregated from multiple mosquito and parasite species (previous models)?

## 2. Material and methods

### (a) Life-history experiment

*Anopheles stephensi* mosquitoes from a long-standing laboratory colony (approx. 40 years) were reared as described in electronic supplementary material, Methods. The life-history experiment was initiated 3 days after adult emergence to permit mating. After being presented with a blood meal for 15 min via a water-jacketed membrane feeder, we randomly distributed 30 host-seeking females into individual cages (16 oz. paper cup; mesh top) to one of six constant temperatures (16°C, 20°C, 24°C, 28°C, 32°C, 36°C ± 0.5°C, 80% ± 5 RH and 12 L : 12D photoperiod). Each individual adult cage contained an oviposition site: a Petri dish secured to the cage bottom containing cotton balls to retain liquid, with filter paper for egg removal and counting. Individuals were offered a blood meal for 15 min each day. Blood meals were scored through visual verification of the abdomen immediately after feeding. Oviposition sites were rehydrated and checked for eggs daily. We followed cohorts of individual females in each temperature until all mosquitoes had died or when less than 7% of the starting population remained. At least two biological replicates were performed at each temperature (*n* = 390).

### (b) Statistical analyses

We used generalized linear mixed models (GLMM) with R package *lme4::glmer()* [30] to estimate the effects of temperature, mosquito age and their interaction on the proportion of females that imbibed blood on a given day (i.e. the number of females that took a blood meal on a given day out of the total number of females alive on that day for each temperature treatment) and the mean daily egg production (i.e. the number of eggs laid on a given day divided by the total number of females alive on that day in a given temperature treatment) (electronic supplementary material, Methods). We used a log-rank test with R package *survival::survdiff()* [31] on Kaplan–Meier estimates to determine if survivorship differed with temperature. Lastly, to determine if the daily survival rate changed across the lifespan of the mosquito, we fit a variety of survival distributions, which allow either for a constant (exponential) or variable daily mortality rate (lognormal, gamma, Gompertz and Weibull) with R package *flexsurv* [32] to the Kaplan–Meier estimates (electronic supplementary material, Methods).

### (c) Temperature-dependent transmission potential (relative $R_0$)

We used a temperature-dependent formulation of relative $R_0$ parameterized from the *An. stephensi*—*P. falciparum* system to (i) evaluate the effect of age-related changes in *An. stephensi* life history on the predicted thermal suitability of *P. falciparum* (i.e. 'observed' versus 'estimated' trait values), and (ii) compare predicted thermal suitability for malaria transmission to a previous expression for relative $R_0(T)$ that aimed to describe the

*An. gambiae*—*P. falciparum* system but consisted of data aggregated from several different mosquito and parasite species [10]. To evaluate relative $R_0$, we rescaled a common expression for $R_0$ to range between 0 and 1, which was derived from the Ross–MacDonald model [18,33], initially expanded on in Parham & Michael [16] to incorporate the effect of temperature and rainfall on mosquito life history and thereby mosquito population size, and later modified in Mordecai et al. [11] to approximate individual lifetime reproductive values using daily fecundity output and adult daily mortality rates as a function of temperature without the effect of rainfall on mosquito abundance. (equation (2.1), electronic supplementary material, Methods and table S5) [6,7,10–16,33]:

$$R_0(T)_{\text{estimated}} = \sqrt{\frac{a^*(T)^2 bc(T)e^{-\mu^*(T)/\text{PDR}(T)}\text{EFD}^*(T)p_{\text{EA}}(T)\text{MDR}(T)}{Nr\mu^*(T)^3}}.$$

(2.1)

$R_0$ is the expected number of new cases generated by a single infectious person or mosquito introduced into a fully susceptible population throughout the period within which the person or mosquito is infectious. $R_0$ components include: egg-to-adult survival probability ($p_{\text{EA}}$), mosquito development rate (MDR), fecundity (EFD; eggs laid per female per day), biting rate ($a$), adult mosquito mortality rate ($\mu$), parasite development rate (PDR), vector competence ($bc$; the proportion of parasite-exposed mosquitoes that become infectious), the density of humans ($n$) and the human recovery rate ($r$), with ($T$) indicating parameters that are dependent on environmental temperature (°C). The host recovery rate ($r$) and host density ($n$) are assumed to be temperature independent. We label the $R_0(T)$ formulation in equation (2.1) as 'estimated' as lifetime traits are commonly parameterized with indirect estimates (denoted by *) based on daily rates [6,7,10–15].

To reproduce the multi-species estimated $R_0(T)$ model (which uses 'estimated' trait values), we used the thermal relationships defined in [10] in equation (2.1). To compare the multi-species estimated model to the $R_0(T)$ model parameterized with our *An. stephensi* data (*An. stephensi* estimated) and using the formulation in equation (2.1), we generated trait estimates (denoted by *) according to methods described in [10,11,34,35] for biting rate ($a^*$), lifespan ($lf^*$ as $1/\mu^*$) and lifetime egg production ($B^*$ as EFD$^*/\mu^*$). Briefly, the inverse of the duration of the first gonotrophic cycle for each individual was used to estimate the biting rate ($a^*$). Exponential curves were fit to the tail of mosquito survivorship distributions as described in [11] to estimate the daily mortality rate ($\mu^*$) of mosquitoes at each temperature treatment, and lifespan was assumed to equal the inverse of daily mortality rate. Daily egg production (EFD$^*$) at each temperature was estimated by dividing the number of eggs laid for each female in her first gonotrophic cycle by the number of days in that gonotrophic cycle. Additionally, to estimate *An. stephensi* mosquito development rate (MDR) and probability of egg to adult survival ($p_{\text{EA}}$), as well as *P. falciparum* development rate (PDR) and vector competence ($bc$), we used data from [36] and [29]. Finally, to incorporate the temperature-dependence of each of the traits outlined above and below, we fit nonlinear responses using Bayesian inference as described in Johnson et al. [10] and electronic supplementary material, Methods.

To determine if $R_0(T)$ for *An. stephensi* varies when directly observed lifetime trait values for biting rate ($a$), lifespan ($lf$) and lifetime egg production ($B$) are incorporated instead of estimates generated from a truncated portion of the lifespan, we generated the following $R_0(T)$ formulation (equation (2.2), electronic supplementary material, Methods and table S5).

$$R_0(T)_{\text{lifetime}} = \sqrt{\frac{a(T)^2 bc(T)\Upsilon(T)B(T)p_{\text{EA}}(T)\text{MDR}(T)lf(T)^2}{Nr}}.$$

(2.2)

Mosquito lifespan ($lf$) was defined as the number of days a mosquito survives after being placed in her temperature treatment. Individual biting rate ($a$) was defined as the total number of blood meals a female imbibes divided by the number of days in her lifespan ($lf$), lifetime egg production ($B$) is defined as the total number of eggs laid by a female during her lifespan ($lf$). The directly observed biting rate ($a$), lifespan ($lf$) and lifetime egg production ($B$) were substituted for the indirectly estimated biting rate ($a^*$), lifespan ($lf^* = 1/\mu^*$) and lifetime egg production ($B^* = \text{EFD}^*/\mu^*$) in equation (2.1). The proportion of mosquitoes surviving the latency period, denoted as $\Upsilon$ in equation (2.2), is substituted for $\exp[-\mu/\text{PDR}]$ in equation (2.1). To estimate $\Upsilon$, we fit a Gompertz distribution to survivorship data from each temperature and replicate. We then took the proportion of mosquitoes alive upon completion of the predicted extrinsic incubation period ($\text{PDR}_{50}(T)^{-1}$) of *P. falciparum* at each temperature. The amount of days to reach 50% of maximum infectiousness in a mosquito population is represented by $\text{PDR}_{50}(T)^{-1}$ [29]. This formulation allows us to account for age-dependent mortality in the proportion of mosquitoes surviving the latency period ($\Upsilon$). We then compared the thermal responses of lifespan, biting rate and lifetime egg production for *An. stephensi* when these traits are directly observed ($lf$, $a$, $B$) versus estimated ($lf^*$, $a^*$, $B^*$) from the data generated in this study, as well as if any observed differences alter the predicted thermal suitability for malaria transmission ($R_0$).

As done previously [6,7,10–15], we use *relative* values of $R_0$, as opposed to absolute values, to estimate temperature suitability for malaria transmission across the current distribution of *An. stephensi* in Southern Asia because absolute values of $R_0$ depend on a number of factors that vary by location and time (e.g. mosquito habitat availability and quality, number of human hosts). By rescaling $R_0(T)$ to a range between 0 and 1, we can easily compare the thermal optimum and limits for relative $R_0$ across all formulations. However, when adopting a relative approach, the stable transmission threshold of $R_0 > 1$ is no longer meaningful. Therefore, a conservative suitability threshold of relative $R_0(T) > 0$ is implemented where temperatures outside of this range are deemed unsuitable for transmission because one or more of the components in $R_0(T)$ is equal to zero. Using this suitability threshold, we generated maps depicting the number of months an area is predicted to be thermally suitable for transmission of human malaria (*P. falciparum*) to illustrate the potential impact differences in the thermal breadth among our relative $R_0(T)$ models have across a relevant landscape (electronic supplementary material, Methods). Finally, sensitivity and uncertainty analyses were performed for our *An. stephensi* models as described in [10] and electronic supplementary material, Methods.

# 3. Results

## (a) Temperature and age shape mosquito traits

A cohort life table study evaluated the effect of temperature on *An. stephensi* life-history traits as individuals age. We found the inclusion of higher-order fixed effects for day and temperature in GLMM models for both the proportion of females that imbibed blood on a given day and mean daily egg production ranked higher than GLMM models which assumed only linear effects of temperature and age (electronic supplementary material, tables S1 and S2). Both temperature and mosquito age significantly affected the proportion of females that imbibed blood on a given day, mean daily egg production and survivorship (figure 1, electronic supplementary material, table S3). Further, the interaction between the nonlinear terms for temperature and age significantly affected the proportion of females that imbibed blood on a given day and mean daily egg production (electronic supplementary material, table S3). The proportion of

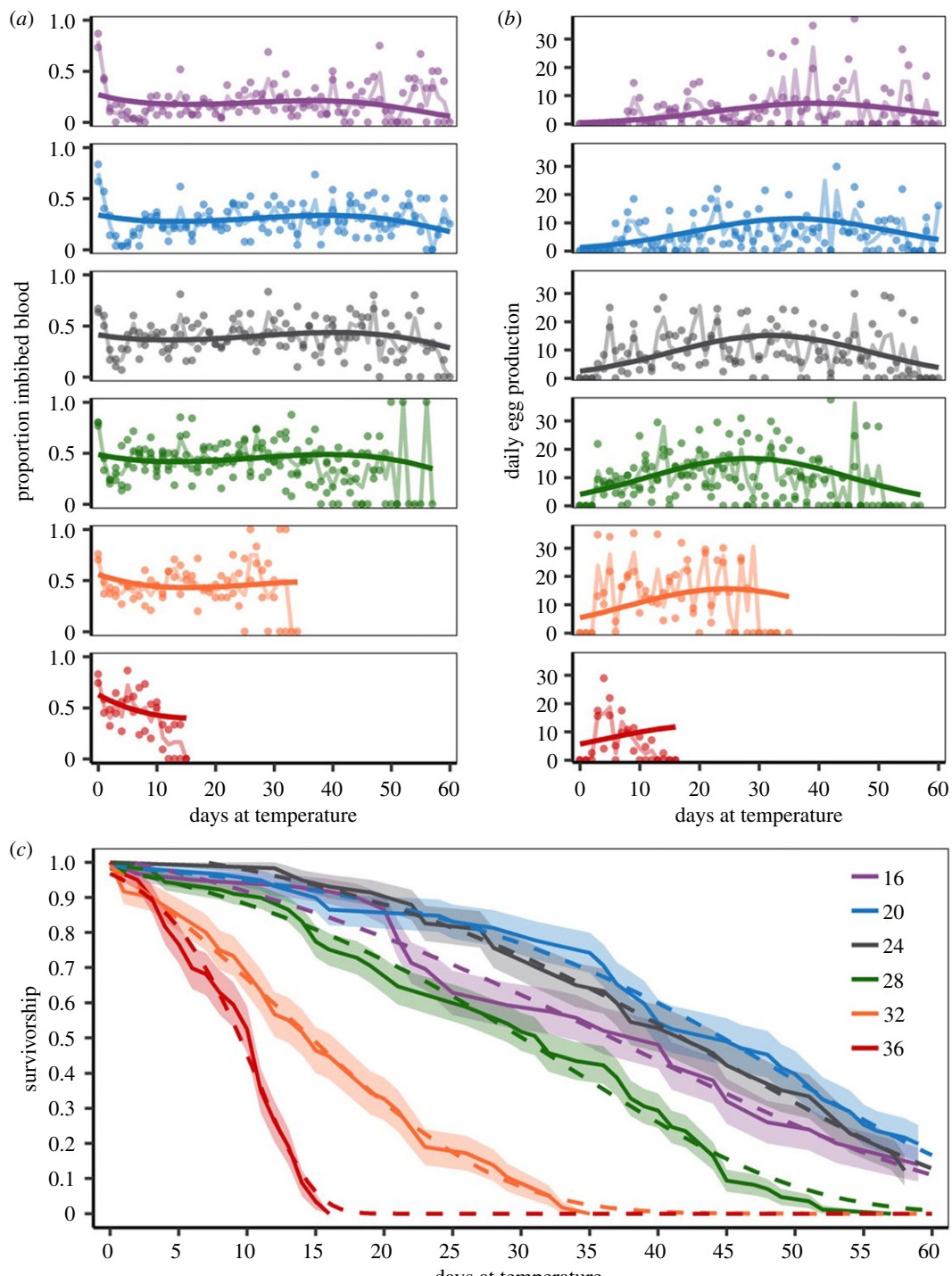

**Figure 1.** Temporal effects. *Anopheles stephensi* life history daily values at 16°C (purple), 20°C (blue), 24°C (grey), 28°C (green), 32°C (orange) and 36°C (red) for the (*a*) proportion of females that imbibed blood, (*b*) daily egg production and (*c*) survivorship. In (*a*) and (*b*), dots represent mean trait values for each replicate, with GLMM model predictions (solid line) and the trend across daily means (faded solid line) shown. In (*c*), Kaplan–Meier estimates (solid line, 95% CI: shaded area) and the best-fitting survival distribution (Gompertz; dashed line) are shown. (Online version in colour.)

females that imbibed blood on a given day was generally higher at warmer temperatures and declined as mosquitoes approached the end of their lifespan in all temperature treatments, with this age-associated decline being most pronounced at 36°C (figure 1*a*). Across all temperature treatments, mean daily egg production increased over time to a peak value before declining (figure 1*b*). Peak mean daily egg production varied with temperature: peak values occurred sooner, and persisted for shorter periods of time at warmer temperatures (28–36°C) compared with cooler temperatures (16–24°C) (figure 1*b*). Temperature also significantly affected survivorship (electronic supplementary material, table S4). Survival responded unimodally to

temperature, with a peak at 20°C and a decline at higher and lower temperatures (figure 1*c*). Finally, at all temperatures, mosquito daily probability of survival was not constant with age: a Gompertz distribution, which allows for a variable daily mortality rate, best fit the survival data at each temperature (figure 1*c*, electronic supplementary material, table S4).

### (b) Using observed as opposed to estimated lifetime trait values alters temperature–trait relationships

Depending on the life-history trait examined, using observed versus estimated lifetime values to fit temperature–trait

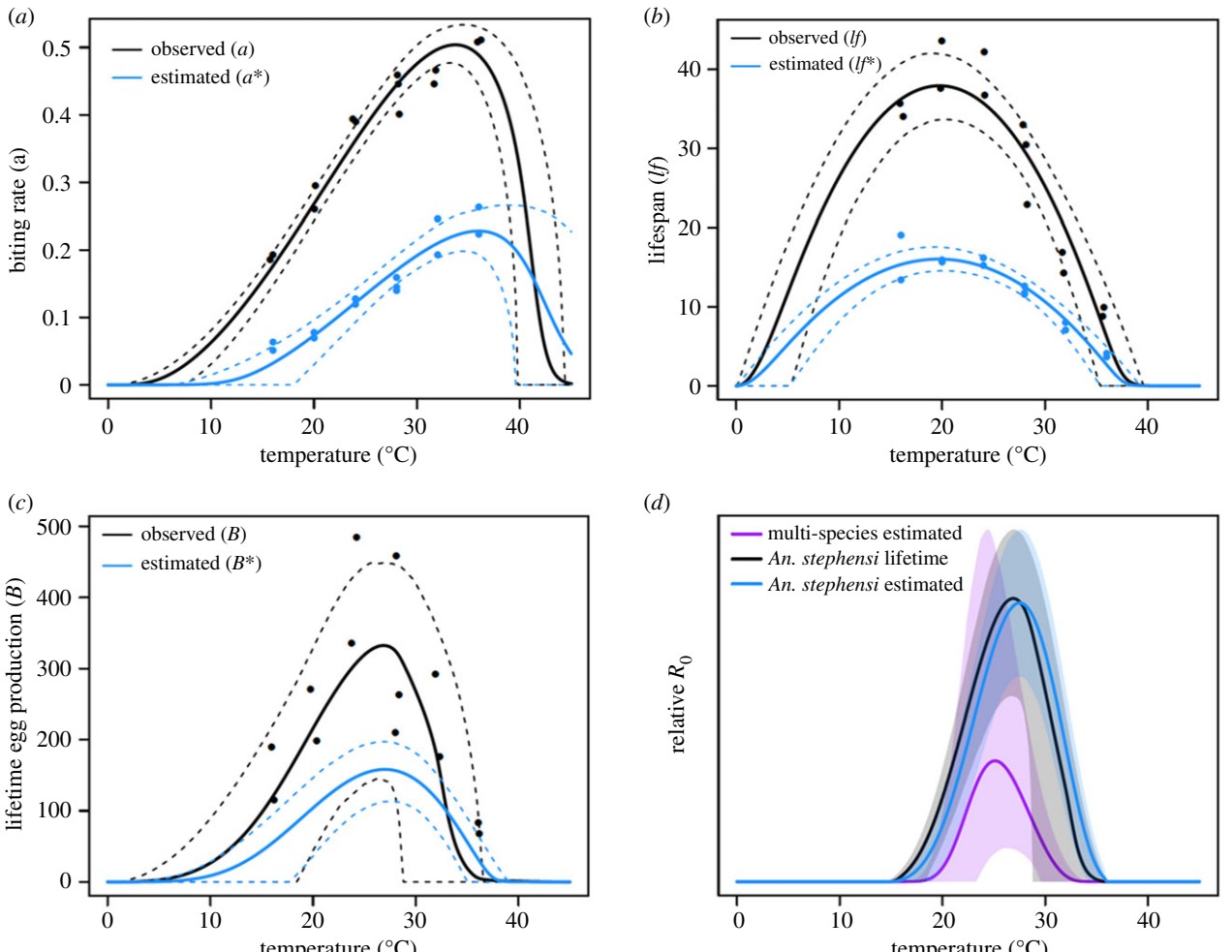

**Figure 2.** Comparison of lifetime and estimated traits. Comparison of temperature–trait relationships between observed lifetime trait values (black) and estimated lifetime values (blue) for (a) biting rate (a), (b) lifespan (lf) and (c) lifetime egg production (B). No data points (dots) are displayed for B* in (c), because this trait is the product of EFD*(T) and lf*(T). (d) Comparison of the three relative $R_0(T)$ models. Each model is plotted relative to the respective max upper 95% CI value with mean model values (solid line) and CI (faded area) across temperature. (Online version in colour.)

relationships resulted in shifts in the predicted thermal minimum ($T_{min}$), maximum ($T_{max}$) and optimum ($T_{opt}$) (figure 2a–c; electronic supplementary material, figure S2, tables S6 and S7). Temperature–trait relationships derived from estimated lifetime trait values resulted in an overall decrease in the absolute values for each trait (figure 2a–c). While peak values of the temperature functions for observed lifetime biting rate (a) were approximately double (0.51 versus 0.24) that of estimated lifetime biting rate (a*), the temperature at which these peak values occurred ($T_{opt}$) was 2.6°C lower for observed lifetime biting rate (figure 2a; electronic supplementary material, table S7). Further, the temperature–trait relationship for estimated biting rate (a*) had a substantially warmer predicted thermal minimum ($T_{min}$; +8.4°C) and moderately higher thermal maximum ($T_{max}$; +1.2°C) than that for lifetime biting rate (a), resulting in a 7.2°C reduction in the breadth of temperatures ($T_{breadth}$) permissive for biting (figure 2a; electronic supplementary material, figure S2a and table S7). Similarly, the value for observed lifespan (lf) at the predicted thermal optimum ($T_{opt}$) was approximately twice that of estimated lifespan (lf*; 38.4 days versus 16.2 days) (figure 2b, electronic supplementary material, table S7). However, in contrast to biting rate, the predicted optimum and maximum temperatures were similar for observed lifespan (lf) and estimated lifespan (lf*) with only a slight 0.2°C

difference in the predicted thermal minimum (figure 2b; electronic supplementary material, figure S2b and table S7). The temperature–trait relationship for observed lifetime egg production (B) was predicted to have a 1.2°C increase in the $T_{opt}$ as compared with estimated lifetime egg production (B*), with a minor 0.4°C increase in the predicted thermal minimum (figure 2c, electronic supplementary material, table S7). Predicted peak values were higher for observed lifetime egg production (B; 396.1 eggs) than estimated lifetime egg production (B*; 175.9 eggs). Finally, there was a major shift in the $T_{max}$ for lifetime egg production between approaches, with directly observed values yielding a $T_{max}$ of 33.2°C and indirect estimates increased the predicted $T_{max}$ (39.8°C) by 6.6°C (figure 2c; electronic supplementary material, figure S2c and table S7).

Surprisingly, the changes in temperature–trait relationships that occurred when observed lifetime data are used instead of estimates did not yield large changes in the predicted relationship between temperature and relative $R_0$ across the *An. stephensi* models. There was a slight decrease in the predicted $T_{opt}$ from 27.6°C (*An. stephensi* estimated) to 27°C (*An. stephensi* lifetime), a moderate increase in the predicted $T_{max}$ from 33°C (*An. stephensi* lifetime) to 35.8°C (*An. stephensi* estimated), but no difference in the predicted $T_{min}$ across models (figure 2d; and electronic supplementary material, figure S3, table S8). As the relative $R_0$ expression varied

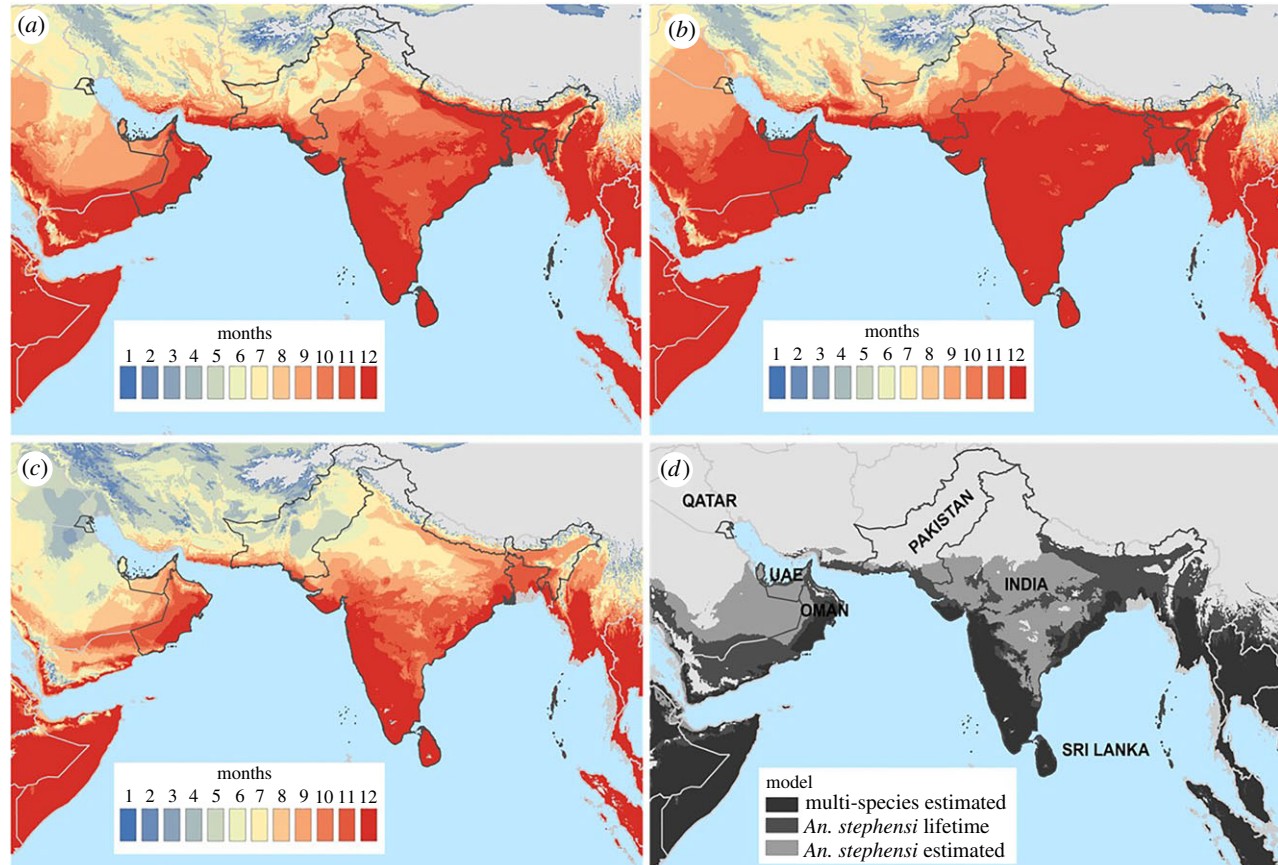

**Figure 3.** Mapping thermal suitability. Number of months with relative $R_0(T) > 0$ for (*a*) *An. stephensi* lifetime, (*b*) *An. stephensi* estimated and (*c*) multi-species estimated traits. (*d*) Mapping overlay of all models with year-round thermal suitability (deep red shading in previous panels) with *An. stephensi* estimated (bottom layer, light grey), *An. stephensi* lifetime (middle layer, dark grey) and multi-species estimated (top layer, black). Thus, the black region corresponds to the areas where all three models predict year-round thermal suitability, the dark grey reflects the additional area predicted by the *An. stephensi* models, while the light grey represents the additional area predicted for the *An. stephensi* estimated model only. (Online version in colour.)

between the *An. stephensi* models, sensitivity and uncertainty analyses were performed to assess the overall contribution of each trait to the resulting fit (electronic supplementary material, Methods). $R_0(T)$ was sensitive to lifespan (*lf*) and biting rate (*a*) in both *An. stephensi* models; however, the *An. stephensi* lifetime model exhibited less sensitivity to lifespan (*lf*) than the *An. stephensi* estimated model (electronic supplementary material, Results, figures S4 and S5). Finally, we found notable differences in the $T_{min}$ and $T_{max}$ of the estimated thermal relationship for the proportion of mosquitoes surviving the latency period ($Y$) between *An. stephensi* models (electronic supplementary material, figure S6 and table S9).

## (c) The relationship between temperature and relative $R_0$ differs from previous estimates

Integrating temperature–trait relationships from the *An. stephensi–P. falciparum* system resulted in a qualitatively different temperature-relative $R_0$ relationship to a previously defined multi-species model [10] (figure 2*d*; electronic supplementary material, figure S3 and tables S6–S8). The *An. stephensi* relative $R_0(T)$ expression parameterized with equivalent trait calculation methods but different trait data (*An. stephensi* estimated) displayed an increase in the breadth of suitable temperatures over which relative $R_0 > 0$ and a decrease in the credible intervals around the thermal minimum ($T_{min}$), maximum ($T_{max}$) and optimum ($T_{opt}$) compared with the multi-species estimated model, which was used to

describe malaria transmission via *An. gambiae* (figure 2*d*; electronic supplementary material, figure S3 and table S8). This increase in temperature breadth results from an increase in $T_{max}$ from 32.4°C (multi-species estimated) to 35.8°C (*An. stephensi* estimated) and a decrease in $T_{min}$ from 19.2°C (multi-species estimated) to 15.6°C (*An. stephensi* estimated). The *An. stephensi* estimated model also had a predicted warmer $T_{opt}$ than the previous multi-species estimated model ($T_{opt}$; 25.6°C) by 2.0°C (figure 2*d*; electronic supplementary material, table S8). In addition, the estimated thermal relationship for the proportion mosquitoes surviving the latency period between the multi-species estimated and *An. stephensi* estimated models differed (electronic supplementary material, figure S6 and table S9).

## (d) Temperature suitability varies geographically across relative $R_0$ models

To visualize differences in model predictions, we created maps illustrating geographical variation in seasonal thermal suitability for *P. falciparum* transmission for each $R_0(T)$ model along with spatial descriptors of the maps across the range of *An. stephensi* (figure 3*a–c*; electronic supplementary material, table S10). Comparisons are drawn to the multi-species estimated model to illustrate how thermal suitability predictions may vary across disease systems. The mapped overlay of year-round (12-months) thermal suitability for malaria transmission highlights the broader geographical

extent of temperature suitability in our *An. stephensi*–*P. falciparum* models, extending northward into India and on the Arabian Peninsula as compared with the previous multi-species estimated $R_0(T)$ model (figure 3*d*; electronic supplementary material, table S10). For example, the multi-species estimated model predicts India to contain 710 046 km$^2$ of temperature-suitable area for transmission year-round and 103 645 km$^2$ in Oman, whereas our two *An. stephensi* models are predicted to contain at minimum approximately double the year-round thermally suitable area based on temperature (electronic supplementary material, table S10). By contrast, the predicted year-round thermally suitable area in Sri Lanka remained largely unaltered among model predictions. Further, Qatar was predicted to be unsuitable for year-round malaria transmission in the multi-species estimated model and *An. stephensi* lifetime model but contained a modest area of year-round temperature suitability (11 210 km$^2$) with our *An. stephensi* estimated model. Lastly, we found a notable increase in temperature suitability in the central regions of India and on the Arabian Peninsula with the *An. stephensi* estimated model (India; 2 372 906 km$^2$) compared with the *An. stephensi* lifetime model (India; 1 352 222 km$^2$) (figure 3; electronic supplementary material, table S10).

## 4. Discussion

Understanding the relative contributions of abiotic and biotic factors to transmission potential is crucial for the prediction and management of infectious diseases. This study characterized how mosquito life-history traits of an urban Indian malaria vector, *An. stephensi*, were jointly modified by temperature and age to affect the temperature suitability for malaria transmission. We found that in addition to temperature, mosquito age altered the daily proportion of females imbibing a bloodmeal, daily egg production and daily probability of survival. These results suggest that estimates of these life-history traits characterized during a finite portion of a mosquito's lifespan may be imprecise [6,10,11]. This study also evaluated how predictions of thermal suitability for malaria transmission are influenced using direct observations for lifespan, lifetime egg production and biting rate instead of common proxies currently used in the literature to estimate these traits. Importantly, we found large quantitative differences in observed lifetime trait values relative to estimates that suggest absolute transmission potential could differ with mosquito age. A failure to include the effects of mosquito age structure could have implications for modelling approaches that predict malaria transmission dynamics. Finally, we determined that the inclusion of *An. stephensi*–*P. falciparum* specific data from either observed lifetime values or commonly used estimates from a truncated portion of the lifespan resulted in qualitatively different temperature–transmission suitability relationships compared with a previous relative $R_0(T)$ model that used thermal responses from *An. gambiae* and other *Anopheles* and *Aedes* species, ultimately affecting predictions of regional thermal suitability for malaria transmission [10,11].

Research across a diversity of ectotherms demonstrates that age and temperature both affect multiple facets of life history [23–25,37–47]. In this study, we observed a decrease in the proportion of females imbibing a blood meal and daily egg production in older *An. stephensi*. For both the proportion of females imbibing blood and survival, the rate of decline occurred faster at increasingly warmer temperatures. Previous work with *An. gambiae* showed an increase in the daily biting rate with age [28], which is in contrast to our findings, yet a separate study found *An. gambiae* biting rates to decrease with gonotrophic cycle and temperature, which is aligned with our results [48]. It remains unclear whether this outcome is due to differences in senescence, allocation of resources or nutritional conditions.

Similar to a previous study, we found daily egg production to decline with age [27]. Further, our data supports that optimal egg laying occurs at moderate temperatures [48]. We found the variation in ages associated with egg laying decreased at higher temperatures where mortality occurred most rapidly (figure 1*b*); however, a previous study observed a negative effect of egg viability at warmer temperatures [48]. Together these data suggest that egg viability is a critical component to be included in temperature-dependent models aimed at predicting population abundances. The temperature-sensitive age-dependent mortality rates for mosquito populations are concordant with previous work in the laboratory and limited field studies [24,49,50]. While there is some evidence that long-lived *An. gambiae* cohorts can occur in the field, it is generally assumed that mosquitoes have shorter lifespans in the field than typically observed in controlled laboratory settings [51,52], and the same may be true for *An. stephensi*.

Finally, there is strong evidence that immune systems senesce in mosquitoes [38] as well as in other organisms [53,54], and that there are age-related changes in mosquito susceptibility to infection [47]. This research suggests the susceptibility of mosquitoes to vector-borne pathogens could change with mosquito age. We did not account for how vector competence and the extrinsic incubation period influence the proportion of the mosquito population that is alive, infectious and biting in our lifetime model, or how these effects scale with the environmental temperature. Thus, whether mosquitoes experience senescence in the field remains an open and critical question [24].

Using direct measurements of an individual's biting rate, lifetime fecundity and lifespan instead of common approaches to estimate these traits from truncated portions of a mosquito's life (e.g. first gonotrophic cycle only) yielded quantitatively, and in some cases qualitatively, different temperature–trait relationships. Our results suggest that previous approaches used to estimate these life-history traits in the literature underestimate values for these traits across most temperatures. This could have important ramifications for predicting mosquito population dynamics including the effect of mosquito control interventions where thermal conditions vary if mosquitoes do experience senescence in the field. Further, imprecise estimates of lifespan can have a compounding effect on predictions of population dynamics and pathogen transmission because lifespan impacts total reproductive output and the amount of time a mosquito is infectious [24,25,55]. More effort (e.g. mark–recapture studies and age-grading technologies) is needed in measuring lifespan and age-associated changes in life history under field settings.

With the relative $R_0$ approach, absolute differences in predicted temperature–trait relationships are masked. Thus, we cannot account for variation in the intensity of malaria transmission with temperature among modelling approaches.

These results suggest that predictions of seasonal prevalence could be improved in a modelling framework that incorporates the age-structure of mosquito populations. By using a relative $R_0(T)$ model we were able to explore how model parameterization of trait data (estimated versus observed) influenced the temperature suitability for *P. falciparum* transmission. While there were substantial quantitative differences between directly observed versus estimated lifetime trait values along with qualitative differences in the shape of the temperature-dependent functions for biting rate and lifetime egg production, we observed minor differences in the thermal response of relative $R_0$ between the *An. stephensi* estimated and *An. stephensi* lifetime models (figure 2*d*).

The subtle shift in the $T_{max}$ between *An. stephensi* relative $R_0$ models resulted in meaningful differences in the predicted thermal suitability of malaria transmission across the known range of the *An. stephensi* vector (figure 3). However, it should be noted that the credible intervals for the *An. stephensi* observed model overlap with both the *An. stephensi* estimated model and the multi-species model at the $T_{max}$, although the density function of the posterior samples at $T_{max}$ suggests these are likely distinct (figure 2*d*; electronic supplementary material, figure S3, tables S6 and S7). In this framework, the limits of predicted thermal suitability are ultimately dictated by which trait has the warmest $T_{min}$ and the coolest $T_{max}$. On the cool end, the probability of egg to adult survival ($p_{EA}$), with the warmest $T_{min}$, constrained both the estimated and lifetime *An. stephensi* models (figure 2*d*; electronic supplementary material, figures S6, S7 and tables S6, S7). By contrast, the traits with the coolest $T_{max}$ values that constrained the predicted temperature-relative $R_0$ relationship differed across *An. stephensi* models (B: lifetime model; $p_{EA}$: estimated model). Further, these traits also dictated the width of credible intervals around relative $R_0(T)$ near the $T_{max}$, resulting in large and small credible intervals associated with the *An. stephensi* lifetime and estimated model, respectively (figure 2*d*; electronic supplementary material, figures S3, S6, S7 and tables S6, S7).

While our *An. stephensi* estimated model was sensitive to lifespan, our *An. stephensi* lifetime model was less so (electronic supplementary material, Results, figures S4 and S5). Thus, the shift in the predicted thermal optimum for relative $R_0$ to cooler temperatures in our *An. stephensi* lifetime model relative to the *An. stephensi* estimated model is largely driven by the qualitative differences in the temperature–trait relationship between observed and estimated biting rate and the proportion of mosquitoes surviving the latency period (figure 2; electronic supplementary material, Results and figure S6). Differences in the temperature–trait relationship for the proportion of mosquitoes surviving the latency period likely arise between models as the *An. stephensi* lifetime model accounts for mortality rates that vary with age, whereas the *An. stephensi* estimated model assumes a constant mortality rate.

Using *An. stephensi* data dramatically changed the predicted relationship between the thermal suitability of malaria transmission and temperature relative to the previously published multi-species estimated model [10], potentially suggesting that the thermal limits and optima of relative $R_0(T)$ models varies across disease systems [5,7]. We demonstrate a 4.3°C decrease in the predicted thermal minimum and 2.6°C increase in the thermal maximum for our *An. stephensi* estimated model, as compared with the multi-species estimated model that used trait responses from multiple *Anopheles* and an *Aedes* species (figure 2*d*;

electronic supplementary material, table S7) [11]. The increase in thermal suitability at warmer temperatures could be due to differences in physiological constraints of the mosquito vectors investigated. *An. stephensi* may be selected for higher temperature tolerance, as it is found in urban areas in Asia. Thus, due to its geographical location and the urban 'heat-island effect,' this species inhabits warmer areas on average than that of the more rural *An. gambiae* [56]. Further, differences in *Plasmodium* species and the method of calculating EIP could drive differences between models [57]. However, this would not explain the increased suitability at cooler temperatures, which instead suggests a vector or parasite with a higher plasticity in temperature tolerance. Finally, incorporating life-history data for *An. stephensi* and *P. falciparum* reduced the credible intervals for all of the predicted thermal thresholds for the temperature-relative $R_0$ relationship relative to the multi-species estimated model, except for the $T_{max}$ associated with the *An. stephensi* lifetime model (figure 2*d*; electronic supplementary material, table S7) [10]. To further refine temperature suitability predictions for effective use in vector control and to optimally inform public health strategies there is a strong need for additional research on temperature effects on the basic biology of disease vectors.

Accurately predicting malaria transmission ultimately depends on additional variation in other abiotic, biotic and socioeconomic factors that determine human exposure to mosquitoes that our relative $R_0$ approach does not capture. For example, it is currently unknown if mosquitoes behaviourally modify their response to temperature in the field. Further, $R_0$ here is static and does not incorporate the effect of temporal variation in daily or seasonal temperatures or fluctuations in vector and host abundances or disease states (i.e. susceptible, exposed, infectious, recovered). Differences in mosquito rearing conditions among laboratories in which the data were generated probably also exist, which could explain some of the differences observed across our two models. Additional study limitations are presented in electronic supplementary material, Discussion. However, this is a fundamental first step in assessing the effect of mosquito age on predicted thermal suitability for malaria transmission, as well as the ability of a previous temperature-dependent model to predict thermal suitability in another relevant mosquito–human malaria system.

In this study, we illustrate that the predicted temperature-relative $R_0$ relationship and land area of thermal suitability were affected by using common approaches to estimate mosquito lifetime traits versus directly measuring them. Further, differences in the overall magnitude of these traits—as opposed to the shapes of their thermal responses—could affect transmission in ways not captured using the relative $R_0(T)$ approach. Lastly, substituting thermal responses with data from *An. stephensi* compared with a previous model which used responses from multiple mosquito species, resulted in substantially more land area predicted to be thermally suitable for year-round malaria transmission in Southeast Asia. This work highlights the importance of careful consideration for how trait values are measured and aggregated into transmission models, and underscores the need for more basic research in the field to improve the accuracy of transmission models.

Data accessibility. Data and associated code are available from the Dryad Digital Repository: https://doi.org/10.5061/dryad.8cz8w9gmd [58].

**Authors' contributions.** K.L.M.—conceived, executed and analysed the data from the presented research, and helped write the manuscript. M.S.S.—assisted with the Bayesian approaches and model fits with the research and helped write the manuscript. S.J.R.—made the maps of our model predictions from the research and helped write the manuscript. O.C.V.—assisted with the Bayesian approaches and model fits with the research and helped write the manuscript. R.J.H.—assisted with the modelling framework and writing the manuscript. J.O., T.A. and K.B.—undergraduate researchers who assisted in the execution of the research. L.R.J. and E.A.M.—consulted on the Bayesian approaches, model fits, the sensitivity and uncertainty analyses, and helped write the manuscript. C.C.M.—contributed to the conception and funding of the project, the experimental design, and writing of the manuscript, and also consulted on the data collection and analysis associated with the research.

**Competing interests.** We declare we have no competing interests

**Funding.** This work was supported in-part by the NSF Graduate Research Fellowship Program and a NIH R01 award (1R01AI110793-01A1). E.A.M., S.J.R., L.R.J. and M.S.S were supported by an NSF EEID grant (DEB 1518681). E.A.M. was also supported by an NIH NIGMS MIRA (1R35GM133439-01), a Hellman faculty fellowship, a Stanford Woods Institute for the Environment—Environmental Ventures Program grant and a Terman Award.

**Acknowledgements.** The authors would like to thank the Center for Undergraduate Research Opportunities and the NSF REU Population Biology of Infectious Diseases program at UGA.

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
