## [Reviewer comments · Proceedings of the Royal Society B: Biological Sciences]

Review History

RSPB-2019-2207.R0 (Original submission)

Review form: Reviewer 1

Recommendation

Major revision is needed (please make suggestions in comments)

Scientific importance: Is the manuscript an original and important contribution to its field?

Good

General interest: Is the paper of sufficient general interest?

Good

Quality of the paper: Is the overall quality of the paper suitable?

Good

Is the length of the paper justified?

Yes

Should the paper be seen by a specialist statistical reviewer?

No

Do you have any concerns about statistical analyses in this paper? If so, please specify them explicitly in your report.

No

It is a condition of publication that authors make their supporting data, code and materials available - either as supplementary material or hosted in an external repository. Please rate, if applicable, the supporting data on the following criteria.

Is it accessible?

No

Is it clear?

No

Is it adequate?

No

Do you have any ethical concerns with this paper?

No

Comments to the Author

I truly enjoyed reading the manuscript by Miazgowicz and colleagues. It is very well written, has a modest number of very clear figures and deals with an important topic in global health research.

Whilst very well written and with very accessible and clearly presented results, there are two major caveats. The authors draw conclusions on the suitability of temperature ranges for Pf transmission and convincingly demonstrate that their experiments have implications on inferences on R0 and transmission. However, temperature and mosquito age are also major factors in determining mosquito susceptibility to plasmodium infections with possible (but unquantified) interactions. This is a major limitation of the current study that ideally would require the relevant experiments or, at a very minimum (not very satisfactory), a clear description in the discussion section. Secondly, the conclusions are drawn on areas with vivax rather than falciparum is the dominant plasmodium species. This should be relatively easily incorporated.

Other specific points:

The history of the mosquito colony is missing. How many generations has the mosquito been in colony and at what temperature has it been maintained? This could affect the validity of the results. A recent colony would in many ways be preferable.

I liked the very clear introduction that ended with 4 clearly defined questions. It would be great if the end of the discussion can return to this structure in a summary statement.

The authors model constant temperature (slightly confusingly referred to as treatment in the figures, I would suggest days at temperature) whilst temperatures of course fluctuate and this could have a profound impact on the examined parameters. Why wasn't an average day and average night temperature inferred? That would greatly have improved the extent to which the experiments mimicked natural conditions.

The authors report the breadth of suitable temperatures over which R0 exceeds 0. Exceeding 1 appears more relevant for sustained transmission.

The authors indicate that it is difficult to examine senescence in field conditions and mention this is a critical question. It would be valuable if they can discuss possible study designs in the (semi-) field.

Whilst very well written and with very accessible and clearly presented results, there are two major caveats. The authors draw conclusions on the suitability of temperature ranges for Pf transmission and convincingly demonstrate that their experiments have implications on inferences on R0 and transmission. However, temperature and mosquito age are also major factors in determining mosquito susceptibility to plasmodium infections with possible (but unquantified) interactions. This is a major limitation of the current study that ideally would require the relevant experiments or, at a very minimum (not very satisfactory), a clear description in the discussion section. Secondly, the conclusions are drawn on areas with vivax rather than falciparum is the dominant plasmodium species. This should be relatively easily incorporated.

Other specific points:

The history of the mosquito colony is missing. How many generations has the mosquito been in colony and at what temperature has it been maintained? This could affect the validity of the results. A recent colony would in many ways be preferable.

I liked the very clear introduction that ended with 4 clearly defined questions. It would be great if the end of the discussion can return to this structure in a summary statement.

The authors model constant temperature (slightly confusingly referred to as treatment in the figures, I would suggest days at temperature) whilst temperatures of course fluctuate and this could have a profound impact on the examined parameters. Why wasn't an average day and average night temperature inferred? That would greatly have improved the extent to which the experiments mimicked natural conditions.

The authors report the breadth of suitable temperatures over which R0 exceeds 0. Exceeding 1 appears more relevant for sustained transmission.

The authors indicate that it is difficult to examine senescence in field conditions and mention this is a critical question. It would be valuable if they can discuss possible study designs in the (semi-) field.

Review form: Reviewer 2

Recommendation

Reject - article is scientifically unsound

Scientific importance: Is the manuscript an original and important contribution to its field?

Marginal

General interest: Is the paper of sufficient general interest?

Acceptable

Quality of the paper: Is the overall quality of the paper suitable?

Poor

Is the length of the paper justified?

No

Should the paper be seen by a specialist statistical reviewer?

No

Do you have any concerns about statistical analyses in this paper? If so, please specify them explicitly in your report.

Yes

It is a condition of publication that authors make their supporting data, code and materials available - either as supplementary material or hosted in an external repository. Please rate, if applicable, the supporting data on the following criteria.

Is it accessible?

Yes

Is it clear?

N/A

Is it adequate?

N/A

Do you have any ethical concerns with this paper?

No

Comments to the Author

This study provides novel insights into the impact of mosquito age and temperature on multiple entomological parameters that influence malaria transmission intensity and is one of the first studies to collect data that simultaneously investigates both these variables in a controlled, laboratory setting. The study could be of interest to both medical entomologists and epidemiologists. There are, however, several statistical considerations and over-interpretation of data which precludes publication in its current form. The relevance of the findings in the field is not considered, and the definition of R_0 ignores the need for mosquito breeding sites and generates results which could be dangerously misleading (please see the report below for your consideration). Many thanks.

Decision letter (RSPB-2019-2207.R0)

11-Nov-2019

Dear Dr Murdock:

I am writing to inform you that your manuscript RSPB-2019-2207 entitled "Mosquito species and age influence thermal performance of traits relevant to malaria transmission" has, in its current form, been rejected for publication in Proceedings B.

This action has been taken on the advice of referees, who have recommended that substantial revisions are necessary. With this in mind we would be happy to consider a resubmission, provided the comments of the referees are fully addressed. However please note that this is not a provisional acceptance.

The resubmission will be treated as a new manuscript. However, we will approach the same reviewers if they are available and it is deemed appropriate to do so by the Editor. Please note that resubmissions must be submitted within six months of the date of this email. In exceptional

circumstances, extensions may be possible if agreed with the Editorial Office. Manuscripts submitted after this date will be automatically rejected.

Sincerely,
Dr Sasha Dall
mailto: proceedingsb@royalsociety.org

Associate Editor

Board Member: 1

Comments to Author:

I have now received two very considered reviews that make a series of detailed criticisms. Both referees worry about how lab adapted this colony is and therefore how relevant these results are to the field. One of the referees is very concerned more broadly about this and makes a number of measured criticisms of the approach and interpretation of the data. The study is limited by using a colony and not looking at infected mosquitoes both in terms of the effect of temperature on infection and transmission but also on temperature dependent mortality. The question becomes how reasonable is it to draw those maps given these limitations? My sense is that this work is hard and important and therefore getting part way to the answer is worth while. We could look at a resubmission, but my sense is that you would need to really address these issues and limitations upfront.

Reviewer(s)' Comments to Author:

Referee: 1

Comments to the Author(s)

I truly enjoyed reading the manuscript by Miazgowicz and colleagues. It is very well written, has a modest number of very clear figures and deals with an important topic in global health research.

Whilst very well written and with very accessible and clearly presented results, there are two major caveats. The authors draw conclusions on the suitability of temperature ranges for Pf transmission and convincingly demonstrate that their experiments have implications on inferences on R0 and transmission. However, temperature and mosquito age are also major factors in determining mosquito susceptibility to plasmodium infections with possible (but unquantified) interactions. This is a major limitation of the current study that ideally would require the relevant experiments or, at a very minimum (not very satisfactory), a clear description in the discussion section. Secondly, the conclusions are drawn on areas with vivax rather than falciparum is the dominant plasmodium species. This should be relatively easily incorporated.

Other specific points:

The history of the mosquito colony is missing. How many generations has the mosquito been in colony and at what temperature has it been maintained? This could affect the validity of the results. A recent colony would in many ways be preferable.

I liked the very clear introduction that ended with 4 clearly defined questions. It would be great if the end of the discussion can return to this structure in a summary statement.

The authors model constant temperature (slightly confusingly referred to as treatment in the figures, I would suggest days at temperature) whilst temperatures of course fluctuate and this could have a profound impact on the examined parameters. Why wasn't an average day and average night temperature inferred? That would greatly have improved the extent to which the experiments mimicked natural conditions.

The authors report the breadth of suitable temperatures over which R_0 exceeds 0. Exceeding 1 appears more relevant for sustained transmission.

The authors indicate that it is difficult to examine senescence in field conditions and mention this is a critical question. It would be valuable if they can discuss possible study designs in the (semi-) field.

Whilst very well written and with very accessible and clearly presented results, there are two major caveats. The authors draw conclusions on the suitability of temperature ranges for Pf transmission and convincingly demonstrate that their experiments have implications on inferences on R_0 and transmission. However, temperature and mosquito age are also major factors in determining mosquito susceptibility to plasmodium infections with possible (but unquantified) interactions. This is a major limitation of the current study that ideally would require the relevant experiments or, at a very minimum (not very satisfactory), a clear description in the discussion section. Secondly, the conclusions are drawn on areas with vivax rather than falciparum is the dominant plasmodium species. This should be relatively easily incorporated.

Other specific points:

The history of the mosquito colony is missing. How many generations has the mosquito been in colony and at what temperature has it been maintained? This could affect the validity of the results. A recent colony would in many ways be preferable.

I liked the very clear introduction that ended with 4 clearly defined questions. It would be great if the end of the discussion can return to this structure in a summary statement.

The authors model constant temperature (slightly confusingly referred to as treatment in the figures, I would suggest days at temperature) whilst temperatures of course fluctuate and this could have a profound impact on the examined parameters. Why wasn't an average day and average night temperature inferred? That would greatly have improved the extent to which the experiments mimicked natural conditions.

The authors report the breadth of suitable temperatures over which R_0 exceeds 0. Exceeding 1 appears more relevant for sustained transmission.

The authors indicate that it is difficult to examine senescence in field conditions and mention this is a critical question. It would be valuable if they can discuss possible study designs in the (semi-) field.

Referee: 2

Comments to the Author(s)

This study provides novel insights into the impact of mosquito age and temperature on multiple entomological parameters that influence malaria transmission intensity and is one of the first studies to collect data that simultaneously investigates both these variables in a controlled, laboratory setting. The study could be of interest to both medical entomologists and epidemiologists. There are, however, several statistical considerations and over-interpretation of data which precludes publication in its current form. The relevance of the findings in the field is not considered, and the definition of R_0 ignores the need for mosquito breeding sites and generates results which could be dangerously misleading (please see the report below for your consideration). Many thanks.

Author's Response to Decision Letter for (RSPB-2019-2207.R0)

See Appendix A.

RSPB-2020-1093.R0

Review form: Reviewer 2

Recommendation

Major revision is needed (please make suggestions in comments)

Scientific importance: Is the manuscript an original and important contribution to its field?

Acceptable

General interest: Is the paper of sufficient general interest?

Acceptable

Quality of the paper: Is the overall quality of the paper suitable?

Acceptable

Is the length of the paper justified?

Yes

Should the paper be seen by a specialist statistical reviewer?

No

Do you have any concerns about statistical analyses in this paper? If so, please specify them explicitly in your report.

No

It is a condition of publication that authors make their supporting data, code and materials available - either as supplementary material or hosted in an external repository. Please rate, if applicable, the supporting data on the following criteria.

Is it accessible?

Yes

Is it clear?

Yes

Is it adequate?

Yes

Do you have any ethical concerns with this paper?

No

Comments to the Author

Many thanks for the improved manuscript. I agree that laboratory studies are currently necessary to separate the effects of temperature from other variables, there are numerous logistical challenges involved and consequently we have to rely on the best methods available at the time. The discussions around these points are improved, thank you. It is nonetheless important to consider the uncertainty involved and extent to which we can draw field-relevant conclusions when interpreting laboratory data. I have added some additional considerations. Thank you for investigating the impact of synchronous biting, modifying Figure 1 and investigating the use of different priors on the temperature-trait relationships – these are improvements.

Further considerations

1. Line 52 – are the “alterations in mosquito behaviour” you are referring to behavioural resistance to insecticide?
2. Lines 197 to 206 – the discussion around the use of a relative or absolute R_0 is a lot clearer now, thank you. I do, however, think it should be defined in the first paragraph of the “Temperature-dependent transmission potential (R_0)” section. Also, this section could be called “Temperature-dependent transmission potential (relative R_0)”. Given the current situation, the use and understanding of the absolute R_0 is now widespread outside the scientific community, avoiding confusion is therefore vital.
3. Lines 291 to 310 – I understand what you mean, but I still feel the findings are being over interpreted.
 - a. The use of the word “endemic” suggests there is year-round transmission, but actually the laboratory mosquito responses suggest that air temperature is suitable year-round for transmission to occur, given a number of assumptions.
 - b. For example, we do not know if/how mosquito behaviour in the field shapes their responses to temperature.
4. The discussion around senescence in the field is improved and important (lines 347 to 351). For further consideration, whilst senescence has been observed in the field if mosquito lifespan is shortened the time over which senescence can impact the observed transmission relevant traits, will be less.
5. The age of the long-standing colony should be included in the main article (not just the supplementary information).

Decision letter (RSPB-2020-1093.R0)

29-May-2020

Dear Dr Murdock:

Your manuscript has now been peer reviewed and the reviews have been assessed by an Associate Editor. The reviewers' comments (not including confidential comments to the Editor) and the comments from the Associate Editor are included at the end of this email for your

reference. As you will see, the reviewers and the Editors have raised some concerns with your manuscript and we would like to invite you to revise your manuscript to address them.

Research ethics:

Use of animals and field studies:

Please submit a copy of your revised paper within three weeks. If we do not hear from you within this time your manuscript will be rejected. If you are unable to meet this deadline please let us know as soon as possible, as we may be able to grant a short extension.

Best wishes,
Dr Sasha Dall
mailto:proceedingsb@royalsociety.org

Associate Editor Board Member

Comments to Author:

Thank you for revising the MS - I really appreciate the work that you have put into the revision and now think this is a really great study. The referee makes some useful points that you should consider in a revision, but I am sure you can address them easily. I really enjoyed reading the paper and think it will make a major contribution.

Reviewer(s)' Comments to Author:

Referee: 2

Comments to the Author(s).

Many thanks for the improved manuscript. I agree that laboratory studies are currently necessary to separate the effects of temperature from other variables, there are numerous logistical challenges involved and consequently we have to rely on the best methods available at the time. The discussions around these points are improved, thank you. It is nonetheless important to consider the uncertainty involved and extent to which we can draw field-relevant conclusions when interpreting laboratory data. I have added some additional considerations. Thank you for investigating the impact of synchronous biting, modifying Figure 1 and investigating the use of different priors on the temperature-trait relationships - these are improvements.

Further considerations

1. Line 52 - are the "alterations in mosquito behaviour" you are referring to behavioural resistance to insecticide?

2. Lines 197 to 206 – the discussion around the use of a relative or absolute R_0 is a lot clearer now, thank you. I do, however, think it should be defined in the first paragraph of the “Temperature-dependent transmission potential (R_0)” section. Also, this section could be called “Temperature-dependent transmission potential (relative R_0)”. Given the current situation, the use and understanding of the absolute R_0 is now widespread outside the scientific community, avoiding confusion is therefore vital.

3. Lines 291 to 310 – I understand what you mean, but I still feel the findings are being over interpreted.

a. The use of the word “endemic” suggests there is year-round transmission, but actually the laboratory mosquito responses suggest that air temperature is suitable year-round for transmission to occur, given a number of assumptions.

b. For example, we do not know if/how mosquito behaviour in the field shapes their responses to temperature.

4. The discussion around senescence in the field is improved and important (lines 347 to 351). For further consideration, whilst senescence has been observed in the field if mosquito lifespan is shortened the time over which senescence can impact the observed transmission relevant traits, will be less.

5. The age of the long-standing colony should be included in the main article (not just the supplementary information).

Author's Response to Decision Letter for (RSPB-2020-1093.R0)

See Appendix B.

RSPB-2020-1093.R1 (Revision)

Review form: Reviewer 1

Recommendation

Accept as is

Scientific importance: Is the manuscript an original and important contribution to its field?

Good

General interest: Is the paper of sufficient general interest?

Acceptable

Quality of the paper: Is the overall quality of the paper suitable?

Acceptable

Is the length of the paper justified?

Yes

Should the paper be seen by a specialist statistical reviewer?

No

Do you have any concerns about statistical analyses in this paper? If so, please specify them explicitly in your report.

No

It is a condition of publication that authors make their supporting data, code and materials available - either as supplementary material or hosted in an external repository. Please rate, if applicable, the supporting data on the following criteria.

Is it accessible?

Yes

Is it clear?

Yes

Is it adequate?

Yes

Do you have any ethical concerns with this paper?

No

Comments to the Author

Thanks for all your hard work with the revisions.

Decision letter (RSPB-2020-1093.R1)

29-Jun-2020

Dear Dr Murdock

I am pleased to inform you that your manuscript entitled "Age influences the thermal suitability of *Plasmodium falciparum* transmission in the Asian malaria vector *Anopheles stephensi*" has been accepted for publication in Proceedings B.

Open Access

Paper charges

Sincerely,

Dr Sasha Dall

Associate Editor:

Board Member: 1

Comments to Author:

Thank you for your thorough revision - I think this is a great paper

Appendix A

Response to Reviewers' Comments:

We would like to thank the reviewers for all of their thoughtful and insightful comments. After incorporating the below suggestions, we think the manuscript has been improved greatly. We hope the revisions meet the reviewers' expectations. Reviewer comments are included in italics, with our responses in normal text. Changed text in the manuscript have been highlighted using track changes in the new version of the manuscript.

Thank you!

Courtney Murdock

Statement to editor:

The authors feel that addressing each of the reviewers' comments listed below has improved the overall quality of the presented manuscript and warrants consideration of the manuscript in its current form for publication. In response to the reviewers, we have revisited the approach used to fit the thermal responses for the mosquito and pathogen traits included in our models. To do this we involved Drs. Johnson, Villena, and Shocket who have originally developed the fitting approach used throughout this manuscript and are now included as co-authors on the manuscript. Our specific modifications to the temperature-trait fits are outlined in the **SI Methods**. As a result of these modifications some of our overall statements regarding the effects of including directly observed lifetime values as opposed to indirect estimates of lifetime values have been altered slightly.

Referee 1:

Comments to the Author(s)

I truly enjoyed reading the manuscript by Miazgowicz and colleagues. It is very well written, has a modest number of very clear figures and deals with an important topic in global health research.

Whilst very well written and with very accessible and clearly presented results, there are two major caveats. The authors draw conclusions on the suitability of temperature ranges for Pf transmission and convincingly demonstrate that their experiments have implications on inferences on R0 and transmission.

We thank Reviewer #1 for their response and have addressed each of the specific points made below. We feel that the incorporation of their feedback has improved the quality of the manuscript.

However, temperature and mosquito age are also major factors in determining mosquito susceptibility to plasmodium infections with possible (but unquantified) interactions. This is a major limitation of the current study that ideally would require the relevant experiments or, at a very minimum (not very satisfactory), a clear description in the discussion section.

This is an excellent point that the reviewer brings up and we agree completely. There is good evidence that immune systems senesce in mosquitoes (1) as well as in other systems (2, 3), which suggests the susceptibility of mosquitoes to vector-borne pathogens could change depending on mosquito age. These effects could have ramifications for our estimates of both vector competence and the extrinsic incubation period influencing the proportion of the mosquito population that is alive, infectious, and biting with the magnitude of these effects varying with environmental temperature. We have now incorporated this into the discussion section (**Discussion** ‘clean’ lines 350-356).

While we recognize the potential importance of these effects on our predicted relative $R_0(T)$, we feel that demonstrating these effects is outside the scope of this manuscript. Investigating the effects of age and infection, as well as how temperature mediates these effects, on $R_0(T)$ could be an entirely separate manuscript. This study is not meant to directly relate to field dynamics but is used as a theoretical framework to aid future field efforts in collecting the type of data required to refine predictive models.

Secondly, the conclusions are drawn on areas with vivax rather than falciparum is the dominant plasmodium species. This should be relatively easily incorporated.

While *An. stephensi* is generally recognized as being the primary vector of *P. vivax*, *An. stephensi* is also capable of transmitting *P. falciparum* in urban areas (4-6) across its current distribution (depicted in our mapping figures), as well as in recent invasion zones in Africa (7-10). We do agree that it would have been good to also include thermal suitability predictions for *P. vivax*, but this is currently not possible. There is a significant lack data to characterize these traits in the literature and no viable *P. vivax in vitro* model to generate these data. Thus, we cannot derive with any confidence temperature-trait relationships for vector competence nor the extrinsic incubation period of *P. vivax* infected *An. stephensi* mosquitoes in our models.

Other specific points:

The history of the mosquito colony is missing. How many generations has the mosquito been in colony and at what temperature has it been maintained? This could affect the validity of the results. A recent colony would in many ways be preferable.

We agree with the reviewer that working with colonies recently derived from the field would be better for the scope of our questions than a long standing colony of *An. stephensi*. However, because of the biology of *Anopheles* as well as the bureaucratic oversight in India, it is virtually impossible to source a colony from the field without doing the work in country. This is currently included as a limitation of our study (**SI_Discussion**), emphasized as a long standing colony in the methods (**Methods** ‘clean’ line 103), and throughout the discussion attempt to draw parallels with how our data relates to the results of other studies conducted in the lab and field (**Discussion** ‘clean’ lines 337-338, 345-349, 350-352, 356-357, 363-365, 368-370). Further, we have provided as much information as we can on the origin, type, etc. of the colony we used in our experiments in the **SI_Methods** of the manuscript.

I liked the very clear introduction that ended with 4 clearly defined questions. It would be great if the end of the discussion can return to this structure in a summary statement.

We used the first paragraph of the Discussion [**Discussion** ‘clean’ lines 310-329] to mirror the structure of the last paragraph of the Introduction and to provide a summary statement regarding each of our original research questions.

The authors model constant temperature (slightly confusingly referred to as treatment in the figures, I would suggest days at temperature) whilst temperatures of course fluctuate and this could have a profound impact on the examined parameters. Why wasn't an average day and average night temperature inferred? That would greatly have improved the extent to which the experiments mimicked natural conditions.

Since we manipulated the mean temperatures mosquitoes experienced, temperature is an experimental treatment. However, we agree with Reviewer 1 that simply using ‘days at temperature’ is potentially less confusing to readers. This change has been adapted in **Figure 1** and throughout the manuscript.

We agree that determining how the temperature – relative R_0 relationship varies under more natural conditions (e.g., diurnally fluctuating temperatures) vs. constant mean temperatures is important, but this question is currently outside the scope of this study. One of our main research goals was to generate a relative $R_0(T)$ model specifically with our *An. stephensi* data to be compared to a previously established model that used data aggregated from multiple parasite and mosquito species (11, 12). To make a fair comparison between models, we had to meet the exclusion criteria of data included in the previously published model – one of which being trait data were measured across multiple constant temperatures. To begin characterizing responses to variable temperature comprehensively, we first need thermal performance curves generated at constant temperatures, which we can then integrate over realistic daily temperature ranges. Further, previous work utilizing temperature-dependent R_0 relationships derived from constant temperature experiments have been validated and shown to explain much of the variation in disease incidence in a given area (12-15)(24-28). However, we agree that varying mean constant temperatures is not necessarily reflective of field conditions that vary diurnally, seasonally, etc. and this variation could alter the thermal responses we are characterizing. In fact, we have another manuscript in preparation that explores this very question. This point has been addressed in the limitations section of the discussion (**Discussion** ‘clean’ lines 429-431) and in the SI (**SI_Discussion**).

The authors report the breadth of suitable temperatures over which R_0 exceeds 0. Exceeding 1 appears more relevant for sustained transmission.

Yes, in the traditional sense of the basic reproductive ratio (R_0), $R_0 > 1$ is more relevant for sustained transmission. However, as we describe in the **Methods** (‘clean’ lines 195-207) and **SI_Methods**, we use a more conservative metric of the thermal suitability for transmission termed “relative R_0 ” here and elsewhere (11-17). Absolute R_0 will depend on a wide variety of additional factors that we do not account for in our models (including habitat availability as Reviewer #2 mentions). Thus, when comparing our relative R_0 models we are simply comparing

differences in the critical temperature thresholds where relative R_0 exceeds 0. We then draw maps of the number of months in which the transmission of human malaria is predicted to be thermally suitable (relative $R_0 > 0$) to illustrate how small shifts in critical temperature thresholds can potentially be biologically relevant across a landscape.

The authors indicate that it is difficult to examine senescence in field conditions and mention this is a critical question. It would be valuable if they can discuss possible study designs in the semi-field.

We addressed this comment by including some suggested study designs (cohort mark recapture methods in the field / semi-field) and age grading technologies that could be used on mosquitoes in the field in the discussion section (**Discussion** ‘clean’ lines 368-370).

Referee 2:

Comments to the Author(s)

This study provides novel insights into the impact of mosquito age and temperature on multiple entomological parameters that influence malaria transmission intensity and is one of the first studies to collect data that simultaneously investigates both these variables in a controlled, laboratory setting. The study could be of interest to both medical entomologists and epidemiologists. There are, however, several statistical considerations and over-interpretation of data which precludes publication in its current form. The relevance of the findings in the field is not considered, and the definition of R_0 ignores the need for mosquito breeding sites and generates results which could be dangerously misleading (please see the report below for your consideration). Many thanks.

We thank Reviewer #2 for their response and have addressed each of the specific points made below.

Scientific accuracy, research methods and report

The work is entirely based on laboratory adapted mosquitoes, which are likely to differ to wild mosquitoes, which adapt to local climates (as noted in the supplementary information). The use of a laboratory strain needs to be emphasized in the title and the point made in greater depth in the discussion. The methods should also state when the strain was colonised.

This point was also highlighted by Reviewer #1. Our response is the same as above where we have added text to both the main and supplemental **Methods** to further emphasize these points. As we describe above, logistical limitations make conducting these experiments directly on field-derived mosquitoes, or even recently derived colonies, excessively difficult. We respectfully disagree that this specific point needs to be emphasized in the title of the manuscript. Other considerations on the manuscript title are addressed below.

The title “Mosquito species and age influence thermal performance of traits relevant to malaria transmission” is not sufficiently informative of what was done in the experiment. The title infers differences between species, but only one species is tested here. Consider revising.

To better reflect the theoretical aspect of this body of work, and the distinction in the comparison between relative R_0 models we have retitled the manuscript as “Age influences the thermal suitability of *Plasmodium falciparum* transmission in the Asian malaria vector *Anopheles stephensi*.”

Mosquito life expectancy in the laboratory is substantially greater than in the field (i.e. this study observed mosquitoes living up to ~60 days when most wild populations typically live on average <10days). This is central to the applicability of the results but is only mentioned in the discussion.

We agree with Reviewer #2 that mosquitoes likely have shorter lifespans than mosquitoes in the field as described in the **Discussion** [‘clean’ lines 347-349], although the research quantifying the age of field populations is sparse due to the difficulty of reliably age-grading field

mosquitoes. We have also commented in the **Discussion** [‘clean’ line 365] that some of the implications of our results are contingent on mosquitoes exhibiting senescence in the field, which evidence suggests does occur despite their shorter lifespans in the field (18, 19). One of the strengths of using the relative R_0 model approach is that it is conservative, and the absolute values of specific traits (e.g. the extended lifespan observed in the laboratory) become arbitrary (as only the temperature at which lifespan is minimized or maximized is relevant). We feel that these temperature thresholds would be relatively conserved (see (20)(36)), although there is likely some variation across mosquito populations. Moreover, investigating mosquito trait thermal responses in the laboratory is the only viable way to understand the direct effect of temperature while fully controlling all other variables that would influence mosquito life history in the field, such as humidity, rainfall, human activities, land use, habitat, etc. Previous research using laboratory-derived thermal performance curves to model mosquito-borne disease transmission, including malaria, has shown strong correspondence with field-observed patterns of transmission (12-15).

Furthermore, malaria-infected mosquitoes are expected to have a reduced lifespan. In the study, all mosquitoes are uninfected, and the impact of malaria infection on mosquito mortality is not considered, which is important given that it is infected mosquitoes that are of interest for malaria transmission (Anderson, Knols, & Koella, 2000).

We agree that malaria infection could have direct and indirect effects on mosquito life history traits, as noted in our response to reviewer #1, and that this could have important implications for our predicted temperature-trait relationships and thermal suitability for malaria transmission. However, we do not think that the existing literature has clearly resolved that infected mosquitoes are expected to have a reduced lifespan with *Plasmodium* infection. The presence/absence, strength and direction of life history trade-offs in Anopheline mosquitoes with *Plasmodium* exposure and infection is still controversial. More recent literature reviews such as (21) have highlighted the conflicting results of infection on mosquito survival. Thus, the effect of infection on mosquito survival appears to be context-dependent (including whether natural mosquito-parasite pairings are used, and the presence of environmental hazards associated with biting behavior) and not an interaction that has been fully characterized.

While the effect of infection on mosquito life history traits may be important, it is outside the scope of this current manuscript. For the line of questions addressed in this study, it was necessary to mirror our experimental design to generate data in an equivalent manner as data that have been utilized in previous models, which have been generated using uninfected mosquitoes. To address this point by the reviewer, we have incorporated this as a study limitation in the main **Discussion** (‘clean’ line 353-356,431) and **SI_Discussion**.

Statistical models (GLM and survival analysis) are fit to the laboratory data to investigate the importance of temperature and age.

GLMs are fit to the daily proportion of females that imbibed blood and the daily per capita number of eggs oviposit, with temperature and day treated as continuous fixed effects. Figure 1(a) indicates that on day zero a relatively high proportion of mosquitoes feed, which rapidly declines the following day (as all mosquitoes have just bloodfed). From this point on feeding

becomes less synchronous. It is therefore possible that the significant decline in feeding as mosquitoes age is just a function of this loss of synchronous biting? The impact of age could then be due to the use of three-day old females, which had previously been deprived bloodfeeding, rather than an effect of age. This requires further investigation given that the loess lines look relatively constant, the p -value is >0.01 , and the sample size at older ages will be lower due to mortality. To verify this, models should be refit excluding the first few days of day (say 6) to make sure this early synchrony isn't driving results.

We agree with reviewer #2 that this warrants closer inspection. To address this concern, we have modified the model used for the proportion of females that imbibed blood to be more representative of the response variable reported. Previously we used 'Feed' as our response variable, which was a binary response (0,1) to which a mosquito had fed on a given day. As stated in the manuscript, we use the proportion of mosquitoes which imbibed blood on a given day. Thus, we modified our model so that the response variable is now represented as a proportion calculated from the number of successes and the number of failures of imbibing blood on a given day for the fixed effects of day and temperature. Furthermore, we explored how the inclusion of higher order terms for day and temperature influenced the ranking of our models and model predictions (SI Table 1, 2). We found that the inclusion of a fixed effects for Temperature, Day, Temperature², Day³, and the interaction between Temperature² and Day³, resulted in the highest ranked model (according to both AICc and log-likelihood (LogLik) values on the full dataset for the proportion of females that imbibed blood (SI Table 1). We then fit the same set of models over a truncated dataset as suggested. We choose to exclude data from days 0-4 instead of Days 0-5 as suggested due to the extent of mortality at the higher temperatures. When using the truncated data set the highest ranking GLMM model (lowest AICc) instead contained fixed effect terms for temperature, day, the interaction between temperature and day, temperature², and day².

In the end, even though the best-fitting models differed the model predictions for the GLMM analysis on the full vs. truncated data set were very similar (see Inset Figure 1), suggesting that the effect of age is not solely due to the initial high feed rates (as age was statistically significant in both cases). Further, as females need to feed soon after emergence, we are not surprised that initial feeding rates were high across all treatments and then became asynchronous over time, likely due to temperature effects on mosquito metabolism. Thus, we decided to include the GLMM analysis on the full dataset in the manuscript (revised Figure 1a, SI Table 1).

Inset Figure 1. The proportion of females that imbibed blood over time. Daily means (faded lines), with the GLMM predictions for the best-fitting model over the truncated dataset (dotted lines), and the GLMM predictions for the best-fitting model over the full dataset (solid lines).

We also reevaluated the GLMM models for daily egg production using the same approach. Again, we felt that using models fit to the full dataset better approximated the observed data (revised **Figure 1b**, **SI_Table 2**).

The coefficients of the best-fit models are not shown and should be included in supplemental table 2.

We agree with reviewer #2. The coefficients for all models are shown in **SI_Tables 1, 2**. However, we have modified **SI_Table 3** (previously **SI_Table 2**) to now contain the coefficients of our best fitting model.

Nowhere in the figures are the model fits shown, which is important for model assessment. Fig 1(a & b) presents a loess line and daily mean values which are uninformative, as the results become meaningless when numbers of mosquitoes become small due to mortality.

We agree with Reviewer #2 that the data trends shown with a loess line do not account for the changing sample size across day due to mosquito mortality. Thus, we have modified **Figure 1a, b** to show the GLMM model predictions instead of the loess line.

When referring to Fig 1, line 235 should be Figure 1(b) and line 240 should be Figure 1(c).

Correct. These changes have been implemented. We thank Reviewer #2 for catching this error.

1.6.A “relative R_0 ” is calculated, with the biting rate, lifespan and lifetime egg production calculated from the collected laboratory data. These were calculated under two scenarios, (1) the parameters are estimated from a proportion of the mosquitoes’ lifespan, and (2) from the whole lifespan. Non-linear functions (quadratic or Brière) were fit to these parameter estimates. In estimating the lifespan from a proportion of the mosquitoes’ life, mortality was estimated from only the tail of the survivorship curve (Fig 1a), by fitting an exponential function, which assumes a constant mortality. Given a constant mortality rate we would expect an exponential decline in survivorship, in the data a relatively straight line is observed, thus potentially indicating the observation of senescence. Selecting the tail, where the mortality is greater could explain the difference in observed and estimated mortalities in Fig 2(b). The impact of selecting an earlier proportion of the mosquitoes’ lifespan to generate mortality estimates is not investigated.

The reviewer makes a good point that what portion of the survival curve is used to estimate daily mosquito mortality will result in different temperature – lifespan relationships, and potentially temperature-relative R_0 relationships for our *An. stephensi* estimated model. To do this we followed the methodology specified in (12) to generate our lifespan estimates (μ^{-1}) as we wanted to compare our resulting relative R_0 estimated model to theirs. Thus, we feel that exploring new methodology for estimating mosquito lifespan is outside the scope of this paper. Further, a major strength of using the relative R_0 approach is that our predictions for thermal suitability are not influenced by these scale differences. Instead, our predictions are shaped by the critical thermal limits (T_{min} & T_{max}), which are very similar between approaches. Thus, we do not feel that the methodology used to generate the estimated lifespan values had a large effect on

our thermal suitability predictions. Finally, we qualify our discussion of implications associated with scale differences between estimated and directly observed lifespan with text in the limitations section of the manuscript to address the concern about the lab environment not translating to the field (**SI_Discussion**).

This is necessary given that there is limited/no evidence of senescence in wild-caught mosquitoes, and suggests that the laboratory environment might be generating results that are not observable in the field. This needs to be better investigated, explained and discussed.

Whether senescence occurs in the field or not is an open question due to the logistical difficulties in assessing mosquito age. Limited, but compelling evidence suggests that senescence 1) can occur rapidly (before or within the time frame for when mosquitoes become infectious (1, 22, 23)), 2) can be impacted by environmental variation across a diversity of ectotherm systems (24), 3) could occur in the field despite their shorter lifespans (18, 19, 25), and 4) can have significant implications for predictions of transmission if ignored (19, 26, 27).

1.6.2 Figure 2 extrapolates beyond the observable range of data and imposes functional forms which have no prior justification. This means that the tails of the distributions are highly uncertain, given the importance of these tails on species distribution this needs to be emphasised more.

Research investigating physiological limits in mosquitoes, or invertebrates in general, is not currently standardized or well-characterized. However, a growing body of literature investigating mosquito thermal responses (in addition to other ectotherm taxa) consistently detect the unimodal and non-linear nature of these trait-temperature relationships. Further, establishing these thresholds experimentally is non-trivial, as these are the physiological limits of performance and are inherently hard to measure. This is especially true for adult traits that are subject to carry-over effects (**SI_Discussion**). Thus, we chose to use conservative values (0°C and 45°C) to constrain our relationships in the absence of direct measurements and allowed the data to ultimately dictate the relationship. However, we do take the reviewer's point. The uncertainty associated with these predicted temperature thresholds is reflected in our credible intervals that characterize each temperature-trait relationship and explored in our sensitivity and uncertainty analysis (**SI_Figures 4, 5**). To address this concern, we added text in the **Discussion** ('clean' lines 400-403, 384-388) to highlight the uncertainty associated with these estimates. We also have emphasized the need for additional basic research characterizing these relationships in general, and in the context of field populations as described in previous responses.

For example, in Fig 2(a) it does not appear that biting rate declines from the data points. In a mosquito population we would expect biting rate to decline due to mosquito mortality, but at the individual mosquito level it is not clear whether a decline in biting rate would be observed.

The authors followed the approach of previous literature to dictate the functional forms imposed over the data and the use of a quadratic or Briere was originally informed by the appearance of the data means, along with the fits used previously to model each as in (11, 13, 14, 28). However, we agree with Reviewer #2 that the use of each functional form should be justified. Thus, we have added an additional table to the supplemental information that compares a

quadratic and Briere fit to the data associated with each trait using DIC values (now **SI Table 6**). As the T_{max} of our relative $R_0(T)$ models would not be constrained by a positive linear relationship for either biting rate or pathogen development rate, we are not concerned about potential ramifications of this choice on our thermal suitability predictions. As we now describe in the Discussion the T_{max} of the *An. stephensi* models are dictated by either the probability of egg to adult survival (p_{EA}) or lifetime egg production (B).

The use of the R_0 term is fine when thinking about the malaria theoretically but it becomes dangerously misleading when it is incorporated into maps such as Figure 3.

We agree that using relative R_0 as a heuristic measure of thermal suitability could be easily confused with absolute R_0 . Throughout the previous version of this manuscript we were very careful to not discuss our results in the context of transmission “risk” and focused the discussion on transmission “potential”. That being said, to further remedy the potential misinterpretation associated with the use of our relative R_0 model to the metric R_0 , we changed the language associated with model predictions and mapping to refer specifically to ‘thermal suitability.’

1.7.1 As I understand it, the formulation of R_0 makes the assumption that mosquito density is determined by the laboratory derived eggs per mosquito and their survival (in the lab). Completely missing from this is the availability of suitable breeding sites, which is widely known to determine mosquito range.

We agree with Reviewer #2 that the availability of suitable breeding sites among many other environmental factors (e.g., nutrition abundance / quality, resource competition, predation, etc.) will ultimately influence mosquito density and distribution and have research in this area in another system (29). As for all other traits in the model, we are isolating the effect of temperature on the life history traits and resulting mosquito abundance. This is precisely the reason that $R_0(T)$ is relative, rather than absolute, because factors such as availability of breeding sites vary across settings and affect mosquito abundance. This point is also mentioned in the **Methods** (‘clean’ line 198) and **Discussion** (‘clean’ 427-431). The benefit of using relative R_0 is that it is a conservative estimate of the thermal suitability for transmission to occur. The objective of the maps was to visually demonstrate the fact that small changes in the relative R_0 -temperature relationship can translate to large spatial differences in predicted thermal suitability (**Methods** ‘clean’ lines 204-207) across the current distribution of *An. stephensi* in Asia (**Methods** ‘clean’ 196-197).

*1.7.2 Figure 3 is titled “Mapping of relative $R_0(T)$ ” which is not the cases as it instead plots the number of months where the temperature range is suitable for *A. stephensi*. In the supplementary information (Derivation of $R_0(T)$ models) the authors note that a “relative R_0 ” is used, that only demonstrates the relative impact of temperature, and should not be interpreted in the traditional sense. The authors acknowledge that previous work has described mosquito density as a function of both precipitation and temperature, however for this work rainfall was dropped. This is an incredibly important caveat and should not be hidden in the SI where most people won’t read it.*

We agree with Reviewer #2 that the title associated with **Figure 3** could be potentially misleading as mentioned in the above comments. We have changed the title of **Figure 3** to

“Mapping of thermal suitability.” To further emphasize that our models are based solely on temperature relationships and do not incorporate other sources of environmental variation such as precipitation, we removed the use of ‘environmental’ associated with our suitability predictions throughout and replaced it with ‘thermal or temperature’ suitability. Further, as the removal of precipitation is an important modification to the modeling of mosquito abundance, we have added text in the main **Methods** [‘clean’ lines 137-142] to make this modification more readily apparent to readers.

1.7.3 Therefore, Figure 3 is not mapping the reproduction number and explains why results suggest malaria should be found in the desert. These are key points which limit the application of the results to malaria transmission in the field and should be made clearer in the main text. Reference to the “relative R0” should be excluded from the abstract as this is not what is being presented and a more appropriate term found and used throughout.

See above responses associated with relative R_0 and thermal suitability. The urban type form of *An. stephensi* breeds in man-made containers and watered habitat (cisterns, wells, overhead tanks, etc.), so if these are provided in the desert there will be breeding habitat for this mosquito vector even when there is limited rainfall. This is precisely why we do not incorporate aridity or rainfall maps overlaid with temperature because it could artificially exclude places where people live, store water, and where vectors could be present. We explicitly use ‘thermal suitability’ when discussing the relative R_0 model outputs in order to clarify this distinction in the main text. Text in the **Abstract** [lines 37-39] was modified to be more explicit in that we are using relative R_0 as a metric for the thermal suitability of transmission. To maintain transparency and consistency with the body of literature that implements this framework we elected to keep the use of ‘relative R_0 .’

1.8 The “relative R0” parameterised with the collected laboratory data is compared to a “relative R0” parameterised from multiple sources (“multi-species model”).

1.8.1 The attribution of differences between the “multi-species model” and “Anopheles stephensi model” to mosquito species is limited by confounding factors, the data used in the “multi-species” model came from different laboratories and the priors used differed in Plasmodium species investigated, with respect to EIP and vector competence. Plasmodium species have been demonstrated to differ in development times (with fast and slow developers) (Vaughan, 2007).

We agree with reviewer #2 that there are several potential issues with aggregating data from different mosquito-parasite systems in the multi-species model. To be fair, there is a general lack of high quality temperature-trait data in general, which is why previous approaches compiled data from a variety of sources to generate malaria thermal suitability predictions. Our study wanted to explore whether or not models that aggregate data across multiple mosquito-parasite systems (generated from different labs, using different mosquito and parasite species / strains) produced similar temperature-transmission relationships for a given mosquito-parasite system (*An. stephensi* – *P. falciparum*), and we find that it does not. It should be noted that the data compiled into our *An. stephensi* models also originates from different research groups including the current study (30, 31); however, the same strain of *An. stephensi* and *P. falciparum*, as well as environmental conditions for rearing, were maintained across these groups. We have built in qualifying language to address this concern in the **Discussion** [‘clean’ lines 431-433], the title of

the result section (changed from “The relationship between temperature and relative R_0 is **disease system specific**” to “The relationship between temperature and relative R_0 **differs from previous estimates**”), and the last sentence of our concluding paragraph (**Discussion** [‘clean’ lines 455–448]) to stress careful consideration of the data chosen to integrate into these types of models.

To address the issue raised about the priors, we revisited our original analysis again and do agree that in the original **SI_Figure 1** panel H for parasite development rate (PDR) did indicate that the use of these informative priors caused a premature decline in the thermal relationship at the warmer end than what we might expect from just using uninformative priors over the data. During the revision process we refit PDR using weaker informative priors (e.g., meaning that the use of priors would not substantially shift the thermal relationship from that observed from the original dataset (see revised **SI_Figure 1**)). This change resulted in the predicted temperature relationship for PDR deviating less from the means of the raw data.

References:

- Anderson, R. A., Knols, B. G. J., & Koella, J. C. (2000). *Plasmodium falciparum* sporozoites increase feeding-associated mortality of their mosquito hosts *Anopheles gambiae* s.l. *Parasitology*, 120(4), 329–333. <https://doi.org/DOI: undefined>
- Craig, M. H., Snow, R. W., & le Sueur, D. (1999). A Climate-based Distribution Model of Malaria Transmission in Sub-Saharan Africa. *Parasitology Today*, 15(3), 105–111. [https://doi.org/https://doi.org/10.1016/S0169-4758\(99\)01396-4](https://doi.org/https://doi.org/10.1016/S0169-4758(99)01396-4)
- Vaughan, J. A. (2007). Population dynamics of *Plasmodium sporogony*. *Trends in Parasitology*, 23(2), 63–70. <https://doi.org/https://doi.org/10.1016/j.pt.2006.12.009>

Response Literature Cited

1. Hillyer JF, Schmidt SL, Fuchs JF, Boyle JP, Christensen BM. Age-associated mortality in immune challenged mosquitoes (*Aedes aegypti*) correlates with a decrease in haemocyte numbers. *Cell Microbiol.* 2005;7(1):39-51.
2. Moret Y, Schmid-Hempel P. Immune responses of bumblebee workers as a function of individual and colony age: senescence versus plastic adjustment of the immune function. *Oikos.* 2009;118(3):371-8.
3. Martin LB, Weil ZM, Nelson RJ. Refining approaches and diversifying directions in ecoimmunology. *Integrative and Comparative Biology.* 2006;46(6):1030-9.
4. Santos-Vega M, Bouma MJ, Kohli V, Pascual M. Population density, climate variables and poverty synergistically structure spatial risk in urban malaria in India. *PLoS Negl Trop Dis.* 2016;10(12):e0005155.
5. Bhadra A, Ionides EL, Laneri K, Pascual M, Bouma M, Dhiman RC. Malaria in northwest India: data analysis via partially observed stochastic differential equation models driven by Lévy noise. *Journal of the American Statistical Association.* 2011;106(494):440-51.

6. Laneri K, Bhadra A, Ionides EL, Bouma M, Dhiman RC, Yadav RS, et al. Forcing versus feedback: epidemic malaria and monsoon rains in northwest India. *PLoS Comput Biol*. 2010;6(9):e1000898.
7. Ashine T, Teka H, Esayas E, Messenger LA, Chali W, Meerstein-Kessel L, et al. *Anopheles stephensi* as an emerging malaria vector in the Horn of Africa with high susceptibility to Ethiopian *Plasmodium vivax* and *Plasmodium falciparum* isolates. *bioRxiv*. 2020:2020.02.22.961284.
8. Seyfarth M, Khaireh BA, Abdi AA, Bouh SM, Faulde MK. Five years following first detection of *Anopheles stephensi* (Diptera: Culicidae) in Djibouti, Horn of Africa: populations established-malaria emerging. *Parasitol Res*. 2019;118(3):725-32.
9. Faulde MK, Rueda LM, Khaireh BA. First record of the Asian malaria vector *Anopheles stephensi* and its possible role in the resurgence of malaria in Djibouti, Horn of Africa. *Acta Trop*. 2014;139:39-43.
10. Takken W, Lindsay S. Increased threat of urban malaria from *Anopheles stephensi* mosquitoes, Africa. *Emerg Infect Dis*. 2019;25(7):1431-3.
11. Johnson LR, Ben-Horin T, Lafferty KD, McNally A, Mordecai E, Paaijmans KP, et al. Understanding uncertainty in temperature effects on vector-borne disease: a Bayesian approach. *Ecology*. 2015;96(1):203-13.
12. Mordecai EA, Paaijmans KP, Johnson LR, Balzer C, Ben-Horin T, Moor E, et al. Optimal temperature for malaria transmission is dramatically lower than previously predicted. *Ecol Lett*. 2013;16(1):22-30.
13. Shocket MS, Ryan SJ, Mordecai EA. Temperature explains broad patterns of Ross River virus transmission. *eLife*. 2018;7:e37762.
14. Tesla B, Demakovsky LR, Mordecai EA, Ryan SJ, Bonds MH, Ngonghala CN, et al. Temperature drives Zika virus transmission: evidence from empirical and mathematical models. *Proc Biol Sci*. 2018;285(1884).
15. Mordecai E, Cohen J, Evans MV, Gudapati P, Johnson LR, Lippi CA, et al. Detecting the impact of temperature on transmission of Zika, dengue, and chikungunya using mechanistic models. *PLoS Negl Trop Dis*. 2017;11(4):e0005568.
16. Ryan SJ, McNally A, Johnson LR, Mordecai EA, Ben-Horin T, Paaijmans KP, et al. Mapping physiological suitability limits for malaria in Africa under climate change. *Vector-Borne and Zoonotic Diseases*. 2015;15(12):718-25.
17. Huber JH, Childs ML, Caldwell JM, Mordecai EA. Seasonal temperature variation influences climate suitability for dengue, chikungunya, and Zika transmission. *PLoS Negl Trop Dis*. 2018;12(5):e0006451.

18. Christiansen-Jucht C, Erguler K, Shek CY, Basáñez M-G, Parham PE. Modelling *Anopheles gambiae* s.s. population dynamics with temperature- and age-dependent survival. *Int J Environ Res Public Health*. 2015;12(6):5975-6005.
19. Ryan SJ, Ben-Horin T, Johnson LR. Malaria control and senescence: the importance of accounting for the pace and shape of aging in wild mosquitoes. *Ecosphere*. 2015;6(9):art170.
20. Lyons CL, Coetzee M, Terblanche JS, Chown SL. Thermal limits of wild and laboratory strains of two African malaria vector species, *Anopheles arabiensis* and *Anopheles funestus*. *Malar J*. 2012;11(1):226.
21. Ferguson H, Read A. Why is the effect of malaria parasites on mosquito survival still unresolved? *Trends Parasitol*. 2002;18:256 - 61.
22. Mulatier M, Porciani A, Nadalin L, Ahoua Alou LP, Chandre F, Pennetier C, et al. DEET efficacy increases with age in the vector mosquitoes *Anopheles gambiae* s.s. and *Aedes albopictus* (Diptera: Culicidae). *J Med Entomol*. 2018;55(6):1542-8.
23. Jones CM, Sanou A, Guelbeogo WM, Sagnon N, Johnson PC, Ranson H. Aging partially restores the efficacy of malaria vector control in insecticide-resistant populations of *Anopheles gambiae* s.l. from Burkina Faso. *Malar J*. 2012;11:24.
24. Bowler K, Terblanche JS. Insect thermal tolerance: what is the role of ontogeny, ageing and senescence? *Biological Reviews*. 2008;83(3):339-55.
25. Harrington LC, Buonaccorsi JP, Edman JD, Costero A, Kittayapong P, Clark GG, et al. Analysis of survival of young and old *Aedes aegypti* (Diptera: Culicidae) from Puerto Rico and Thailand. *J Med Entomol*. 2001;38.
26. Bellan SE. The importance of age dependent mortality and the extrinsic incubation period in models of mosquito-borne disease transmission and control. *PLoS ONE*. 2010;5(4):e10165.
27. Styer LM, Carey JR, Wang JL, Scott TW. Mosquitoes do senesce: departure from the paradigm of constant mortality. *Am J Trop Med Hyg*. 2007;76(1):111-7.
28. Mordecai EA, Caldwell JM, Grossman MK, Lippi CA, Johnson LR, Neira M, et al. Thermal biology of mosquito-borne disease. *Ecol Lett*. 2019;22(10):1690-708.
29. Evans MV, Hintz CW, Jones L, Shiao J, Solano N, Drake JM, et al. Microclimate and larval habitat density predict adult *Aedes albopictus* abundance in urban areas. *The American Journal of Tropical Medicine and Hygiene*. 2019;101(2):362-70.
30. Shapiro LLM, Whitehead SA, Thomas MB. Quantifying the effects of temperature on mosquito and parasite traits that determine the transmission potential of human malaria. *PLoS Biol*. 2017;15(10):e2003489.

31. Paaijmans KP, Heinig RL, Seliga RA, Blanford JI, Blanford S, Murdock CC, et al. Temperature variation makes ectotherms more sensitive to climate change. *Global Change Biology*. 2013;19(8):2373-80.

Appendix B

Response to Reviewers

Associate Editor Board Member

Comments to Author:

Thank you for revising the MS - I really appreciate the work that you have put into the revision and now think this is a really great study. The referee makes some useful points that you should consider in a revision, but I am sure you can address them easily. I really enjoyed reading the paper and think it will make a major contribution.

We thank the associate editor board member for their comments. Our responses to the reviewer comments are in italic text and changes made to the manuscript are indicated by track changes. To address the additional comments in the main text while fulfilling the length requirements for Proc B, some additional text was rephrased to be more concise and is also indicated by track changes.

Reviewer(s)' Comments to Author:

Referee: 2

Comments to the Author(s).

Many thanks for the improved manuscript. I agree that laboratory studies are currently necessary to separate the effects of temperature from other variables, there are numerous logistical challenges involved and consequently we have to rely on the best methods available at the time. The discussions around these points are improved, thank you. It is nonetheless important to consider the uncertainty involved and extent to which we can draw field-relevant conclusions when interpreting laboratory data. I have added some additional considerations. Thank you for investigating the impact of synchronous biting, modifying Figure 1 and investigating the use of different priors on the temperature-trait relationships – these are improvements.

Further considerations

1. Line 52 – are the “alterations in mosquito behaviour” you are referring to behavioural resistance to insecticide?

Yes. Specifically, we were referring to field studies which have noted a shift in peak feeding times for Anopheles species in heavily insecticide treated areas (1) which may render the use of bed nets less effective as a contact barrier and as an adulticide. To clarify this point we have added the above citation immediately following this statement in the main text [line 50].

2. Lines 197 to 206 – the discussion around the use of a relative or absolute R_0 is a lot clearer now, thank you. I do, however, think it should be defined in the first paragraph of the

“Temperature-dependent transmission potential (R_0)” section. Also, this section could be called “Temperature-dependent transmission potential (relative R_0)”. Given the current situation, the use and understanding of the absolute R_0 is now widespread outside the scientific community, avoiding confusion is therefore vital.

We agree that avoiding confusion between these closely related terms is essential. Thus, we have modified the Method section heading to be “Temperature-dependent transmission potential (relative R_0)” as suggested by the reviewer [line 130]. Further, we have added ‘relative’ before R_0 [line 131] and modified the sentence on lines 137-138 to define more clearly ‘relative R_0 ’ in the first paragraph of this section. The revised sentence with modifications underlined now reads; “To evaluate relative R_0 , we rescaled a common expression for R_0 to range between 0 and 1, which was derived from the Ross-MacDonald model (17,32), initially...”.

3. Lines 291 to 310 – I understand what you mean, but I still feel the findings are being over interpreted.

a. The use of the word “endemic” suggests there is year-round transmission, but actually the laboratory mosquito responses suggest that air temperature is suitable year-round for transmission to occur, given a number of assumptions.

To be considerate of the important distinction between “endemic” and “thermally-suitable for transmission to occur year-round” we have removed the use of “endemic” when discussing the mapping products and instead directly refer to them as ‘thermally-suitable for year-round transmission’ throughout [lines 299, 301, 302, 305, 447, 480, 484], in the SI_Methods, and in SI_Table 10.

b. For example, we do not know if/how mosquito behaviour in the field shapes their responses to temperature.

The reviewers are correct in that we do not know if mosquito behavior in the field would modify their thermal response from that which is measured in a laboratory setting. We have added a sentence [lines 430-431] to address this limitation. “For example, it is currently unknown if mosquitoes can behaviorally modify their response to temperature in the field.”

4. The discussion around senescence in the field is improved and important (lines 347 to 351). For further consideration, whilst senescence has been observed in the field if mosquito lifespan is shortened the time over which senescence can impact the observed transmission relevant traits, will be less.

While shortened mosquito lifespans in the field may lessen the impact of senescence on transmission relevant traits, it is similarly unclear if senescence could be accelerated in the field compared to in the laboratory where mosquitoes cope with additional stressors (e.g., predation) that are not present in a controlled setting. Thus, we simply emphasize the need for additional research in the field [lines 357-358, and concluding sentence line 449] to refine the impacts of mosquito age on transmission relevant traits.

5. The age of the long-standing colony should be included in the main article (not just the supplementary information).

We have added “(~40 years)” to the methods in the main text [line 103] to indicate the age of the colony.

References

1. Cooke MK, Kahindi SC, Oriango RM, Owaga C, Ayoma E, Mabuka D, et al. 'A bite before bed': exposure to malaria vectors outside the times of net use in the highlands of western Kenya. *Malar J.* 2015;14:259.